ecology, computational biology

climate change, circuit theory, Condatis, connectivity, dispersal, range shifts

**Authors for correspondence:**
Thomas J. P. Travers
e-mail: t.travers@liverpool.ac.uk
Jamie Alison
e-mail: jalison@ceh.ac.uk

†These authors should be considered joint first authors with equally important contributions to the study.

## PUBLISHING

# Habitat patches providing south–north connectivity are under-protected in a fragmented landscape

Thomas J. P. Travers[1,†], Jamie Alison[2,†], Sarah D. Taylor[3], Humphrey Q. P. Crick[3] and Jenny A. Hodgson[1]

[1]Department of Evolution, Ecology, and Behaviour, University of Liverpool, Crown Street, Liverpool, Merseyside L69 7ZB, UK
[2]UK Centre for Ecology and Hydrology, Environment Centre Wales, Deiniol Road, Bangor LL57 2UW, UK
[3]Natural England, York YO1 7PX, UK

TJPT, 0000-0002-3899-7032; JA, 0000-0002-6787-6192; HQPC, 0000-0002-5136-378X; JAH, 0000-0003-2297-3631

As species' ranges shift to track climate change, conservationists increasingly recognize the need to consider connectivity when designating protected areas (PAs). In fragmented landscapes, some habitat patches are more important than others in maintaining connectivity, and methods are needed for their identification. Here, using the Condatis methodology, we model range expansion through an adaptation of circuit theory. Specifically, we map 'flow' through 16 conservation priority habitat networks in England, quantifying how patches contribute to functional South–North connectivity. We also explore how much additional connectivity could be protected via a connectivity-led protection procedure. We find high-flow patches are often left out of existing PAs; across 12 of 16 habitat networks, connectivity protection falls short of area protection by 13.6% on average. We conclude that the legacy of past protection decisions has left habitat-specialist species vulnerable to climate change. This situation may be mirrored in many countries which have similar habitat protection principles. Addressing this requires specific planning tools that can account for the directions species may shift. Our connectivity-led reserve selection procedure efficiently identifies additional PAs that prioritize connectivity, protecting a median of 40.9% more connectivity in these landscapes with just a 10% increase in area.

## 1. Introduction

Species can be hampered in their ability to shift ranges as an adaptation to climate change [1] where there are synergistic negative impacts of anthropogenic land use [2,3]. We need to safeguard species' ability to respond to climate change by incorporating regional and national connectivity into conservation planning [4]. Many studies look at how easily individuals can traverse landscapes [5,6], but modelling landscape connectivity across one or few generations is unlikely to predict long-term, large-scale responses to climate change. Studies need to assess multi-generational connectivity, i.e. whether there is enough habitat in the right places to facilitate long-distance range shifts. Landscape-scale decision-making is crucial to deliver climate-resilient landscapes [7], and losing habitat patches from critical regions between current and projected ranges will hamper species' range expansion—potentially causing extinction [8].

Recently, we have seen a global shift towards promoting functionally connected networks, typified by Aichi biodiversity target 11 [9]. National examples of this include the UK Government's plan to develop a nationwide Nature Recovery Network to protect, restore and connect the country's wildlife

sites [10]. Such initiatives cause stakeholders to reconsider where to prioritize conservation of priority habitats. For both pragmatic and strategic reasons, conservation may have historically favoured larger patches over small ones, thus avoiding fragmented regions [11,12]. However, there remains active debate on the value of several small patches for species richness, versus one contiguous patch of the same size [13,14]. Simulations of species persistence and expansion, using simple metapopulation models, highlight the strengths and weaknesses of different habitat creation strategies. In general, aggregation strategies are good for facilitating metapopulation persistence but not for range expansions, because large gaps are left between habitat aggregations in the direction of range advance [15,16].

Safeguarding habitats in protected areas (PAs) is a widespread, cost-effective tool for biodiversity conservation [17]. Many studies have demonstrated the representation of species' projected future ranges in existing PA networks [18]. Others have shown existing PAs may facilitate species' range expansions by supporting high abundances of, and preferential colonization by, range expanding species [19]. However, colonization does not necessarily lead to successful range expansions, and an important subset of species are failing to shift their range. If protection was lost in patches critical to reaching the projected range, even more species could be vulnerable, and up to now, the protection of such critical patches has generally not been prioritized. Following intensive research, software can now incorporate connectivity in relation to climate change into the decision-making process [7,20–22]. Work is ongoing to put connectivity science into practice, and incorporate connectivity in a nuanced, 'climate-wise' context [7,23]. To that end, tools to identify and protect habitat patches that are crucial for range expansion need to be developed and disseminated.

Successful inclusion of connectivity in conservation decision-making also depends on legal and ecological context. In England, a 2006 Act of Parliament [24] provides for the conservation of listed priority habitats and species [25]. Specifically, legally recognized priority habitats, from lowland meadows to blanket bog, are platforms to protect *ca* 1000 priority species. Priority habitats are ecologically distinct from one another, providing for unique subsets of priority species including threatened and specialist plants, fungi, birds, beetles, butterflies, moths and several other taxa [26]. Some species depend on multiple priority habitats, but protection, restoration and conservation decisions are likely to consider each habitat individually. Beyond priority habitats, conservation practice in England now emphasizes building a 'coherent and resilient ecological network', and ensuring that wildlife sites are 'joined up' [27]. During contemporary climatic warming, South–North range shifts have been widely documented in England among many species [26]. In recent decades, those species undertaking range shifts have disproportionately colonized PAs, highlighting PAs' key role in protecting habitats—even in species' potential future ranges [28].

Here, we use connectivity analysis to inform decision-making within the constraints of a specific policy context. We assess the capacity of England's PAs to secure long-distance connectivity in 16 national conservation priority habitat networks. We define habitat networks as assemblages of patches of a given priority habitat type, because priority habitats receive distinct legal recognition and underpin

planning decisions in England, and are highly ecologically distinct, providing for unique subsets of priority species. We use Condatis [29,30], a landscape-scale decision-support software, to identify habitat patches (i.e. contiguous clumps of habitat) critical to long-distance connectivity and range expansion under climate change. Condatis uses circuit theory to efficiently calculate how quickly a species could reach a specified target location from a specified source. It has mathematical similarities to, but key conceptual differences from, the circuit theory models used by other landscape ecologists [31]: one link in the Condatis network represents a population sending colonists to an empty patch to found a new population (not a disperser stepping between one cell and its neighbour). Crucially, if a patch in Condatis has high 'flow', it is located on one of the likeliest routes for range expansion between the source and the target [30].

To better understand and conserve priority habitats under climate change, we ask: (i) to what extent are high-flow habitat patches represented in England's current PA network? (ii) How is the protection and/or high-flow status of habitat patches related to their area? (iii) How much extra network connectivity could be conserved through targeted conservation of high-flow habitat patches? We use generalized linear models to explore relationships between the flow, size and protection status of patches across priority habitat networks. We rank unprotected habitat patches based on their contribution to long-distance connectivity, and strategically add them to the PA network to demonstrate how targeted conservation could efficiently increase connectivity for a given increase in PA coverage.

## 2. Methods

### (a) Data preparation

Spatial data for the Priority Habitat Inventory (PHI), Sites of Special Scientific Interest and National Nature Reserves (henceforth collectively PAs) in England were downloaded from the Natural England Open Data Geoportal [32]. Polygons of England were downloaded from the Ordnance Survey OpenData Boundary-Line Layer [33].

The PHI represents a broad range of semi-natural habitat types identified as the most threatened and requiring conservation action under the UK Biodiversity Action Plan. These data were all originally in vector format. The PHI polygons were converted to a 50 m raster using ArcMap 10.6, with cell values corresponding to the habitat type of the polygon their centroid intersected. Where cells intersected polygons of multiple habitat types, the rarest took precedence. We merged (i) upland and lowland calcareous grassland habitat types and (ii) upland heathland, lowland heathland, and mountain heathland and willow scrub habitat types (table 1) because of functional similarity between them. The minimum mapping unit of the PHI is 0.1 ha, while the raster resolution equates to 0.25 ha. Therefore, it is unavoidable that a small number of habitat patches will have been lost in the rasterization process. However, we consider it unlikely to be so prevalent that it significantly influenced the findings (mean area lost = 1.8%; electronic supplementary material, table S2).

To consistently represent the colonization process across both large and small patches, Condatis works best with a raster of habitat cells at the finest resolution that will not overwhelm the RAM available (more information in electronic supplementary material, appendix SB). For the 16 habitat networks in our

**Table 1.** Habitats initially included in the study in descending order of area.

| habitat | code | area (ha) |
| --- | --- | --- |
| deciduous woodland | wood | 736 511 |
| heathland[a] | heath | 285 475 |
| blanket bog | blbog | 230 950 |
| coastal floodplain grazing marsh | marsh | 217 556 |
| calcareous grassland[b] | cgrass | 71 075 |
| mudflats | mudfl | 61 261 |
| salt marsh | saltm | 34 111 |
| lowland meadows | lmead | 21 174 |
| lowland fens | lowfens | 20 294 |
| traditional orchard | orchard | 16 023 |
| lowland dry acid grassland | agrass | 15 179 |
| maritime cliff and slope | cliff | 13 348 |
| coastal sand dunes | dunes | 10 227 |
| upland flushes, fens and swamps | upfens | 10 005 |
| purple moor grass and rush pastures | pastures | 9105 |
| lowland raised bog | lrbog | 7814 |
| coastal vegetated shingle | shingle | 3985 |
| reedbeds | reeds | 3136 |
| upland hay meadow | hay | 2439 |
| saline lagoons | lagoons | 1360 |
| limestone pavement | pavement | 1268 |
| calaminarian grassland | calam | 297 |

[a]Heathland network formed of lowland heathland (56 418 ha), upland heathland (227 646 ha), and mountain heaths and willow scrub (1411 ha).
[b]Calcareous Grassland network formed of lowland (61 856 ha) and upland (9219 ha) calcareous grassland.

study, the feasible analysis resolution was 2 km for deciduous woodland due to its large extent (table 1), and 1 km for all other priority habitat types. Thus, habitat cells for the Condatis network (as defined in the next section) were derived by aggregating a 50 m resolution raster using the 'rgdal' [34] and 'raster' [35] packages in R 3.5.0 [36], converting the sum of 50 m habitat cells to a proportional cover.

## (b) Condatis analysis

Condatis is a conservation decision-support tool that adapts circuit theory to predict the speed at which a population could expand its range through a habitat network [29,30]. Range shifts in response to climate change are likely to occur over distances far greater than an individual would be able to traverse in a single lifetime. Unlike other uses of circuit theory, Condatis models multi-generational movements by accounting for reproduction within the breeding habitat, producing successive waves of emigrants.

In the Condatis analogy, each landscape cell containing breeding habitat becomes a node in the circuit network (cells from the rasters described in the previous section). The time taken for a breeding population to colonize one cell from another becomes the resistance between the two. A resistance link is placed between every habitat cell and every other. The matrix outside breeding habitat is assumed to be homogeneous, through which the population can move, but cannot breed, meaning that matrix cells do not form part of any Condatis calculations.

The reciprocal of resistance between habitat cells $i$ and $j$—the colonization rate—is calculated as

$$p_i p_j R \cdot \frac{\alpha^2}{2\pi} \cdot \exp(-\alpha d_{ij}), \qquad (2.1)$$

where $p$ is the area of habitat in each cell, $R$ is the reproductive rate, $2/\alpha$ is the mean dispersal distance, $d$ is the distance between cells $i$ and $j$. The $p$ and $R$ values determine the number of dispersers leaving and arriving in the cells. The distribution of dispersers declines with distance according to a negative exponential kernel.

This simplified dispersal process is a considerable assumption. The way in which we calculate resistance does not model the expected difficulty of moving through the matrix and means we cannot represent physical barriers to dispersal as models like Circuitscape can (see electronic supplementary material, appendix SA for further explanation of these different applications of circuit theory). However, the benefit is that we can analyse much more extensive networks without hitting computational limits, and efficiently represent the long-term, range-level process of range expansion, whose success depends both on reproduction within, as well as dispersal between, habitat cells [29].

Having converted the landscape to a resistor network, we must define source locations—where the population starts—and target locations—where range expansion is deemed successful. A voltage gradient applied from the source to the target causes current to flow—predominantly through the routes of lowest resistance.

Circuit theory calculations lead to an evaluation of the overall connectedness of the habitat network (defined by the metric 'conductance'—a property of the entire network), and the relative importance of each cell to the overall landscape connectivity (defined by the metric 'flow'—a property of individual habitat cells), from a multi-generational dispersal perspective. See electronic supplementary material, appendix SB for details of how conductance and flow are calculated. Conductance is strongly correlated to the speed with which simulated metapopulations can reach the target from the source over multiple generations and all possible travel 'routes' [29]. Flow is a good indicator of the reduction in connectivity that would occur if the habitat cell was deleted from the landscape [30].

## (c) Condatis settings

We ran Condatis for each of the 16 priority habitat networks, and three exemplar mean dispersal distances. We did not attempt to make exact species-specific predictions; instead, we focused on habitat networks as platforms for conservation actions, using traits and processes relevant for multiple species. For the mean dispersal distance trait ($2/\alpha$; equation (2.1)), 2, 4 and 8 km options were run, aiming to represent a broad range of plants, fungi, vertebrates and invertebrate species specialized to each priority habitat network. While many relevant species likely have dispersal abilities of less than 2 km, Condatis calculations encountered rounding errors if the average dispersal was several orders of magnitude lower than the largest gap in the network. Reproductive rate ($R$; equation (2.1)) was fixed at 100 throughout, equating to the production of one emigrant per hectare. This was not based on specific data but is plausible for a medium-bodied vertebrate, or an invertebrate with a low population density. Varying $R$ would not have affected the relative performance of networks and patches, which were the focus of this study, because $R$ modulates all flow and conductance values in proportion.

We identified sources and targets for Condatis on the premise that species are adjusting their ranges to higher latitudes

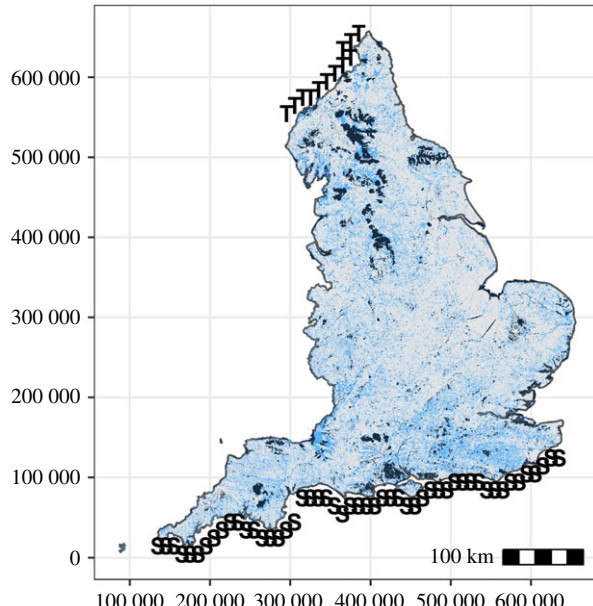

**Figure 1.** Habitat distribution, and source and targets. Spatial distribution of all habitats used in the analysis defined as protected (dark) and unprotected (light). Source (S) and target (T) cells used as an input to Condatis. Coordinates correspond to the Ordnance Survey (OS) British National Grid (measured in metres). (Online version in colour.)

[1]. Thus, a 10 km raster file was produced with sources along the south coast of England, and targets along the northern border with Scotland (figure 1).

### (d) Patch flow and protection assignment

For each habitat network and dispersal option, results were returned as a raster of flow across habitat cells at 1 km resolution (2 km for deciduous woodland). Protection decisions are normally made for habitat patches. Therefore, within each 50 m habitat raster, we identified patches as contiguous clumps of grid cells that share an edge and/or vertex. These patches were assigned flow values of the 1 or 2 km habitat cell they intersected. Where a habitat cell contained more than one patch, flow was divided in proportion to the patches' areas (electronic supplementary material, figure S1). Then, for patches that intersected multiple habitat cells, flow assignments were summed. A geometric average of flow was taken for each habitat patch across the three analysed dispersal distances. The rank of each patch in terms of flow (its 'flow rank') was taken to represent its importance to connectivity. Finally, each patch was classified as 'protected' if more than 50% of its area was covered by PAs. The resulting dataset included protection status and the flow rank of each habitat patch across a range of dispersal abilities.

### (e) Statistical analysis

All statistical analysis was conducted using R 3.5.0 [36]. Graphics and maps were produced in R using '*ggplot2*' [37]. Linear regression analysis was performed to investigate the relationship between log-transformed total habitat network area and conductance.

Comparison of protected and unprotected patches was completed for each habitat network through generalized linear modelling using a binomial distribution, including log-transformed area and flow rank as covariates. Prior to inclusion in the model, flow rank was standardized and centralized. The relationship between patch size and flow was analysed using Kendall rank-order correlations.

The degree of fragmentation of each habitat network was assessed using the GISfrag metric [38]. More contiguous patches, with large amounts of interior habitat, would have had high values, representing a low degree of fragmentation.

To investigate the impact of flow-led patch selection on connectivity protection, we imagined three different protection investment levels: a 1, 10 and 25% increase in the proportion of each habitat network that is protected. Unprotected habitat patches were ranked by flow before being added to the PAs in descending order (highest flow first) until each of the three imagined protection investment levels were met.

## 3. Results

The networks of priority habitat in England range in extent from greater than 0.7 M ha (deciduous woodland) to 297 ha (calaminarian grassland), cover 13.1% of England's land (1.7 million ha total; table 1) and are highly fragmented (median patch size 0.75 ha; electronic supplementary material, table S2). Six habitat networks (salt marsh, maritime cliff and slope, coastal sand dunes, coastal vegetated shingle, saline lagoons and reedbeds) were spatially distributed such that they could not be analysed as electrical circuits at the scales and resolutions used in the study (see electronic supplementary material, appendix C). Of the remaining habitat networks, those covering a larger area facilitated significantly faster speeds of range expansion (regression of log conductance on log-transformed area; $\beta = 3.655$, 95% CI [1.371, 5.940], $R^2 = 0.371$, $F_{1,14} = 9.493$, $p = 0.0073$; figure 2a). Habitat networks also varied widely in the extent to which they are currently protected, ranging from 0.3% (traditional orchard) to 94% (mudflats), with a mean of 53.5% (figure 2b). Although the majority of habitat area was protected in most of the habitat networks (figure 2b), most patches were unprotected (electronic supplementary material, table S2). This was possible because within each habitat network protected patches were, on average, larger than unprotected patches (overall protected mean area 20.98 ha ($n = 32\,253$); overall unprotected mean area 3.58 ha ($n = 287\,737$); electronic supplementary material, table S2), and tended to be less fragmented (protected GISfrag = 258.84, unprotected GISfrag = 84.84; electronic supplementary material, table S2). Proportionally, protection of flow was generally lower than the protection of area; in 12 of the 16 habitat networks, flow protection was, on average, 13.6% lower than area protection. The proportion of flow protected matched or exceeded the proportion of area protected in the remaining four habitat networks (blanket bog (+5.28%), traditional orchard (+0.01%), lowland raised bog (+2.19%) and upland hay meadow (+4.60%); figure 2b).

Larger patches of a given habitat network generally had higher flow. The Kendall rank-order correlations showed weak-to-moderate positive correlations between patch size and patch flow in most habitat networks (electronic supplementary material, table S2; overall $\tau = 0.309$). However, small patches can contribute disproportionally to connectivity; there is wide variation in patch flow values among patches with low area (figure 3a; electronic supplementary material, figure S2). Of the top 10% of patches for flow in each habitat network, an average of 13.8% were patches with an area of less than or equal to 1 ha (electronic supplementary material, table S3).

Given a tendency for larger patches to have higher flow, and to be more often protected, we might expect flow to be well protected. Two results help to show why this is not the

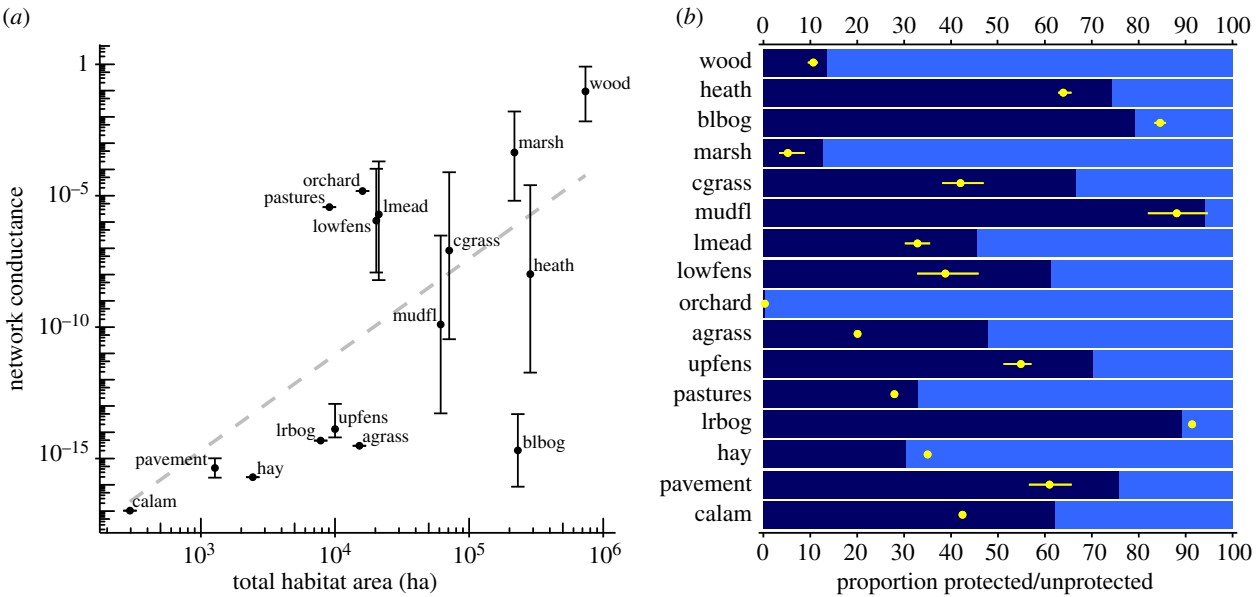

**Figure 2.** Area, conductance and protection levels of 16 priority habitat networks in England. (*a*) Relationship between total habitat area and network conductance, with linear regression (dashed line). (*b*) Proportion of habitat area protected (dark bar) and the proportion of total flow that is protected (point) in each habitat. Points represent geometric means across up to three modelled dispersal distances, while error bars show the range. Habitats are arranged in descending order of total area. (Online version in colour.)

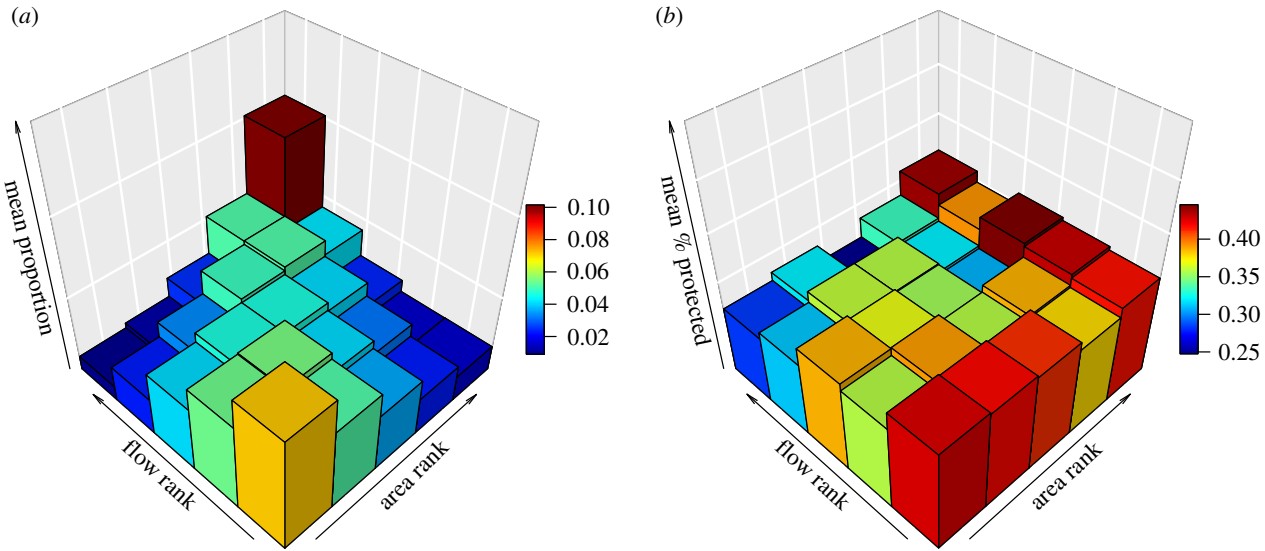

**Figure 3.** Area–flow relationship and its protection. Rank correlations between patch area and flow; ties are assigned random rank. (*a*) Five-by-five three-dimensional histogram (red = high, blue = low) showing the mean proportion of patches, across all habitats, falling into each bin. (*b*) Ranked patch area against ranked flow, showing the mean proportion of patches in each bin that are protected (red = high, blue = low), across all habitats. Habitat by habitat plots are available in the electronic supplementary material. (Online version in colour.)

case. Firstly, among patches that have below-average area, protection level clearly declines with the increasing flow (figure 3*b*; electronic supplementary material, figure S3). Secondly, GLMs that include flow as a predictor of protection indicates that flow has generally negative effects, and those that include both area and flow as predictors of protection status show even more negative effects of flow (figure 4; electronic supplementary material, table S4). Effects of patch area in these GLMs tend to be positive and to become more positive when flow is included as a predictor. Just three habitat networks are exceptions, where the model shows positive effects on protection attributed to both the predictor's area and flow (figure 4; electronic supplementary material, table S4).

In our scenarios in which additional high-flow habitat patches were protected, increases in the proportion of flow protected were almost always greater than increases in the proportion of habitat

area protected (figure 5). In a few cases, disproportionate improvements to overall flow protection were not possible due to insufficient unprotected high-flow patches: specifically, coastal floodplain grazing march (when adding 1, 10 and 25%), blanket bog (adding 10%) and lowland raised bog (adding 10%). However, most connectivity conservation gains were highly disproportionate to the areas of habitat selected for protection. Across all habitat networks, increasing the coverage of PAs by 1, 10 and 25% resulted in respective median increases of 8.0, 40.9 and 57.8% flow protection (figure 5*b*).

## 4. Discussion

Here, we highlight that the connectivity of the fragmented networks studied is vulnerable because patches critical for

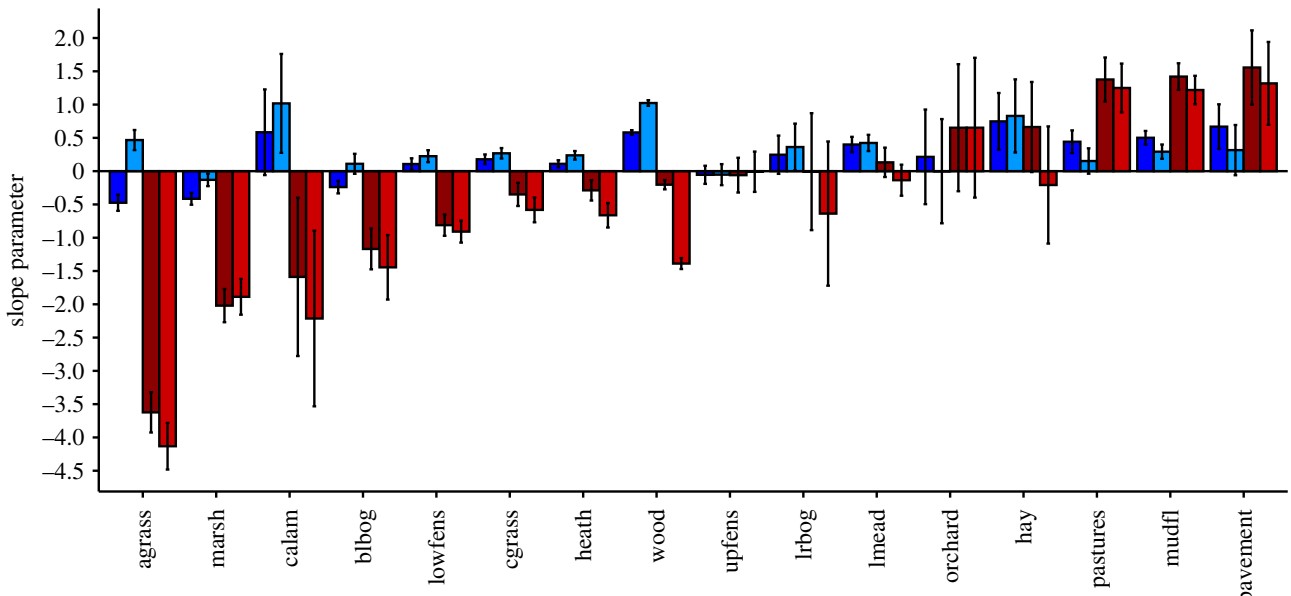

**Figure 4.** Estimates of the effects of patch flow (red) and area (blue) on protection. Parameters for the effect of area and flow on protection in isolation (dark), and together in the same model (light). Habitats are presented in order from lowest to highest flow parameter. Parameter estimates for each habitat area derived from generalized linear models. Error bars represent 95% CIs. (Online version in colour.)

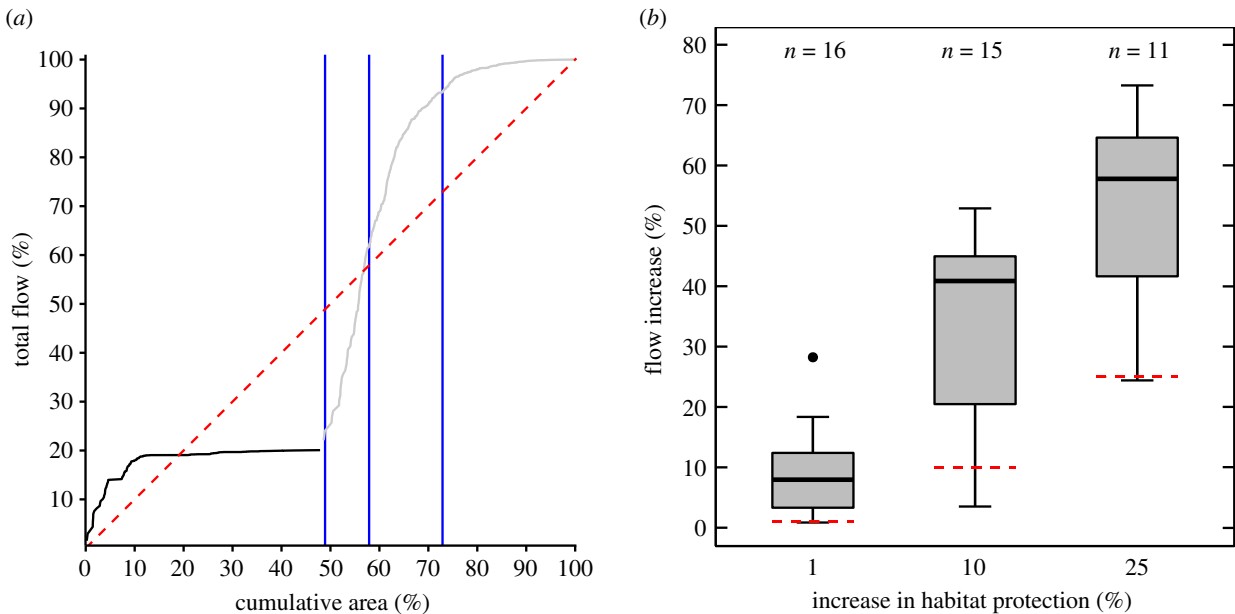

**Figure 5.** Flow protection increases after flow-led patch selection. (*a*) Proportion of flow protected against area of habitat protected (black), and the resulting increase in flow conserved after adoption of flow-led patch selection (grey) for lowland dry acid grassland (chosen as the highest respondent to 25% increase in protection), with data sorted by patch flow in descending order, 1 : 1 trend line (dashed) and vertical lines indicating 1%, 10% and 25% increases in area. (*b*) The increase in flow protection for a 1%, 10% and 25% increase in PA for all habitats after adopting flow-led patch selection. Proportional increase denoted by dashed line, outliers calculated as 1.5 × IQR. (Online version in colour.)

species range expansions are under-protected. Crucially, we found that, for the majority of habitat networks, protection is biased away from high-flow patches (figure 4); the median rate of protection of patches in each habitat network is 44.5% but drops to 37.5% when considering only patches above the 90th percentile for flow. This absence of designation increases the likelihood of degradation or destruction of habitat patches, which is expected to severely impact network connectivity.

Previous research has established that for the majority of countries, PA connectivity is lacking [39]. However, species will not directly respond to PA connectivity *per se*; it is the connectivity of the entire habitat network, whether or not

protected, which affects the reproduction and dispersal of species, and is critical for range expansion under climate change [19]. Unlike previous work, we investigate patch connectivity and patch protection independently. Thus, we contribute ecological realism by focusing not on PA connectivity, but the connectivity of the habitat networks that PAs conserve. We identify the important routes a wide variety of species may take, using simplified dispersal assumptions, as they shift ranges from South to North in reaction to climate change, regardless of protection status (electronic supplementary material, figure S4). In this way, we identify a critical oversight in the design of England's PA network.

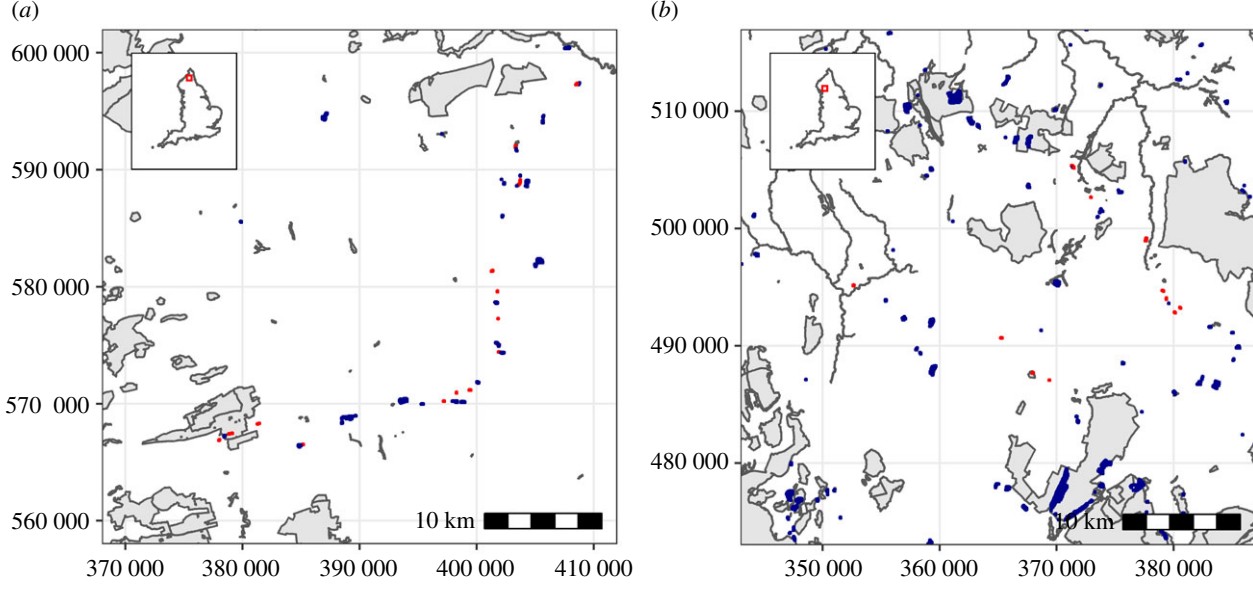

**Figure 6.** Small patches acting as stepping stones. Small habitat patches with flow values in the top 10% (red) positioned in such a way that they act as stepping stones between other areas of habitat (blue) and protected areas (grey), in (*a*) calcareous grassland and (*b*) lowland fens. Coordinates correspond to the Ordinance Survey (OS) British National Grid (measured in metres). (Online version in colour.)

Patches that happen to be strategically located to act as South–North stepping stones (figure 6) may be small and may lack other attributes that were important for past PA designation. We found that existing PAs tended to be biased towards low-flow patches for most habitat networks, despite also being biased towards large patches. This is surprising because large patches typically have higher flow (figure 3*a*). The preferential protection of large patches over small is not a new finding [11]. However, that those same patches typically contribute more to connectivity, and yet connectivity is still under-represented, indicates a disconnect between past protection decisions and those needed to facilitate range shifts.

The patterns we observe in protection are probably not unique to England, given similar biases and lack of PA connectivity have also been evidenced in other regions [12,40]. We propose that comparable network vulnerabilities elsewhere probably result from similar habitat protection principles—and practical considerations—to those known in our study region [41]. For example, reserve selection might actively favour aggregation because, under a stable climate, species persistence is expected to be higher in aggregated networks than fragmented ones [42]. However, passive processes could also be at play. In the UK, many PAs arose from 'Rothschild's Reserves' [27], the selection criteria of which included 'areas of land … which retain primitive conditions and contain rare and local species liable to extinction' [43]. This led to reserves being clustered in areas of low economic and agricultural development, especially in the North and the uplands [44], a phenomenon not limited to the UK [41]. Furthermore, while climate change was not an issue of the time, it is unlikely any form of connectivity was a factor in historical designation decisions, given the growth of PAs in the UK has often occurred without consideration of their wider context [45].

We do not envisage contribution to connectivity—represented here as flow—to be the sole criterion to prioritize protection. Patches that contribute little to connectivity are often crucial to sustain metapopulations [46]. However, we

argue that flow should form part of a nuanced prioritization process, accounting for land-use changes, habitat quality, climate suitability and landscape connectivity [44]. Nevertheless, considering the extent to which high-flow areas have been overlooked, it would not be unreasonable to ring-fence some future PA resources to specifically promote connectivity. Note particularly that flow distribution across patches is highly skewed (on average 31.2% of patches contained 75% of the flow), so future selection of high-flow patches by chance, or by a moderately correlated proxy such as area, is unlikely. By contrast, targeted patch selection on the basis of flow could be very efficient. For example, between 2014 and 2019, terrestrial PAs in the UK increased by 11 200 ha [47]; our analyses show that the addition of 714.25, 438.50 and 3544.50 ha to lowland dry acid grassland, purple moor grass and rush pasture, and calcareous grassland PAs (representing 5% increases in PA) would yield 15.6, 33.1 and 33.5% gains in flow protection, respectively. Such increases in connectivity protection are an urgent requirement if we are to help build more resilient networks for nature in the face of climate change [48].

As connectivity ascends the conservation agenda, we demonstrate the potential for efficient conservation of climate-resilient landscapes. We show that the inclusion of a connectivity measure into the planning process can facilitate the identification of patches important to climate change connectivity, resilience and adaptation. In most habitat networks studied here, substantial gains in connectivity protection can be made for relatively small increases in PA coverage (figure 5*b*). Only for a small number of habitat networks were proportional connectivity gains less than the proportional increase in area. In these instances, either a strong correlation existed between area and flow or existing protection coverage was high, such that the majority of high-flow patches were already protected (electronic supplementary material, table S3). Saura *et al*. [39] identified that the targeted designation of PAs to enhance connectivity was many countries' most pressing priority for meeting PA conservation goals. The flow metric described here provides

potential for proactive safeguarding of connected habitats and stepping stones, allowing conservation planners to target their designation and conservation activities to achieve substantial increases in connectivity protection. This could help to meet targets outlined in legislation such as the UK's 25-year environment plan [10] and the EU's biodiversity strategy, or international commitments, including the upcoming Post-2020 global biodiversity framework [49].

Our study uses cutting-edge methods to quantify protection of long-distance, multi-generational habitat connectivity. Our approach has limitations, but also clear avenues for progression. For example, we analyse 16 priority habitat networks individually, based on the UK Biodiversity Action Plan. Some species are of course reliant on multiple habitat types, and to differing extents, so future work could analyse composite networks of associated habitats used by different subsets of generalist species. However, a more comprehensive assessment might not show any additional crucial patches that had been missed in analyses of individual habitats. Furthermore, if actions increased connectivity for individual priority habitat networks, the connectedness of the composite networks they form part of would also improve. The assumption of a homogeneous matrix is another limitation and may lead Condatis to overestimate the importance of some regions for those species that are hindered by landscape barriers. However, this assumption reduces the computational burden of evaluating connectivity, which is itself a major limitation [50], while still maintaining the principles of isolation by resistance [51]. Another limitation is that while our choice of sources and targets follow the general trend of species moving away from the equator, it does not consider that climate refugia may be found at higher altitudes or different aspects. Furthermore, although the negative exponential kernel at the core of our analysis has been tried-and-tested for modelling animal movements, it may be a poor function for plant dispersal; for example, some studies suggest that log hyperbolic secant or exponential power probability density functions would be more appropriate [52]. Finally, due to our focus on habitats, the scale at which most conservation actions happen, we made use of theoretical species. It would be beneficial to validate our findings empirically with data from species that have already shifted ranges. However, data are not always available, while conservation guidance is needed immediately; many range shifts are ongoing, or have yet to start [53].

Our study quantifies how South–North connectivity is currently conserved within PAs across fragmented habitat networks, using England as an exemplar for application to other countries or regions. Although PAs tend to contain larger patches, which usually contribute more flow, they under-represent connectivity in the majority of habitats studied. The scientific community has been emphasizing the importance of incorporating connectivity into the planning process for at least 30 years [54], but the connectedness of habitats remains vulnerable to degradation and loss. We have shown that patches important to long-distance connectivity can be easily identified, allowing the proportion protected to be greatly increased with minimal additional resources. The decision-making tools demonstrated here help enable the change in conservation planning needed to protect the permeability of landscapes, allowing species to track changing climate and preventing extinction.

Data accessibility. All processed data and R code (including demonstration of Condatis and proportional flow assignment) are available via the Environmental Information Data Centre at https://doi.org/10.5285/a715112e-08ae-4d6e-943a-77933fd5ddd1 [55].

Authors' contributions. T.J.P.T.: conceptualization, data curation, formal analysis, methodology, writing—original draft, writing—review and editing; J.A.: conceptualization, data curation, formal analysis, methodology, writing—original draft, writing—review and editing; S.D.T.: conceptualization, writing—original draft, writing—review and editing; H.Q.P.C.: conceptualization, writing—original draft, writing—review and editing; J.A.H.: conceptualization, data curation, formal analysis, methodology, writing—original draft, writing—review and editing. All authors gave final approval for publication and agreed to be held accountable for the work performed therein.

Competing interests. We declare we have no competing interests.

Funding. T.J.P.T. was funded by an NERC studentship through the ACCE (Adapting to the Challenges of a Changing Environment) Doctoral Training Partnership (grant no. NE/L002450/1), and by Natural England. J.A. and J.A.H. were supported by a Memorandum of Agreement between Natural England and the University of Liverpool. Condatis development was funded by NERC grants (grant nos. NE/L002787/1 and NE/R009597/1).

Acknowledgements. We are grateful to the University of Liverpool Computational Biology Facility, John Heap and Anthony McCabe for High Performance Computing infrastructure and help. We would like to thank two anonymous reviewers whose comments greatly improved the quality of our manuscript.

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
