## [Peer Review File · Proceedings of the Royal Society B: Biological Sciences]

Review History

RSPB-2020-3213.R0 (Original submission)

Review form: Reviewer 1

Recommendation

Major revision is needed (please make suggestions in comments)

Scientific importance: Is the manuscript an original and important contribution to its field?

Acceptable

General interest: Is the paper of sufficient general interest?

Acceptable

Quality of the paper: Is the overall quality of the paper suitable?

Marginal

Is the length of the paper justified?

Yes

Should the paper be seen by a specialist statistical reviewer?

No

Do you have any concerns about statistical analyses in this paper? If so, please specify them explicitly in your report.

No

It is a condition of publication that authors make their supporting data, code and materials available - either as supplementary material or hosted in an external repository. Please rate, if applicable, the supporting data on the following criteria.

Is it accessible?

Yes

Is it clear?

Yes

Is it adequate?

Yes

Do you have any ethical concerns with this paper?

No

Comments to the Author

This is an interesting study, using an adaptation of circuit theory to add nuance to connectivity analyses that are relevant for climate change impact analyses. In general, I appreciated the addition of more biology to the circuit approach, which has several limitations for biological interpretation. I offer the following remarks to strengthen this contribution.

I found the logic supporting the methods and model design choices to be insufficient. The methodological choices don't seem to match the research questions. More logic and explanation is required to make the leap to experimental design. Right now, it seems the approach was designed and tested, then a question was built around the results.

Habitat and patches are not defined. The species that are relevant to this analysis are not mentioned until the conclusions. It is unclear why the authors chose their habitat dataset, nor why they split it up into so many habitat types. If the intent is a broad-scale, multi-species perspective on PAs, I don't see how this approach really does this. I'd expect to see an analysis that runs on a map that integrates all land-cover types. If highlighting specific land-cover types is a key objective, that should be made explicit and explained.

Introduction

38 – synergistic seems an overstatement in a general statement. Additive or coinciding may be true, impacts are not always interacting, multiplicative. Adding qualifying language like “Can” is likely more general and appropriate.

44-48 – repetitive with respect to the multi-generational statements. Condense.

50. Whilst – replace with synonym as meaning is unclear.

72- “as within the Maya.. to Berryessa project in CA”. Suggest you omit or elaborate to describe this project and how it demonstrates your point. There is too little information in the excerpt to be meaningful.

76-77 – For what kind of species have S-N shifts have been documented? This is a good place to indicate what kind of species your analysis will represent (or be relevant for) – plants, terrestrial animal, mammals, birds?

78 – How is habitat defined, if a species-specific view is not used? Suggest omitting ‘habitat’ and replacing it with a more general term.

82 – How is a patch defined?

Methods

93-116. Move after methods/Condatis overview (or integrate within).

153 – unclear.

Results

170. It is still unclear to my why you chose this dataset as representative of habitat. What about species that use more one type of landcover? Why break this apart into types? This doesn't seem needed based on your stated research questions.

196 – Replace 'approaches' with a synonym.

227 – omit: 'we offer a cost-effective remedy'. This is overstated.

228-239: The reads more like introduction than discussion.

254 – if Aichi target is now defunct, why mention it in the introduction? Perhaps substitute another example.

Review form: Reviewer 2

Recommendation

Major revision is needed (please make suggestions in comments)

Scientific importance: Is the manuscript an original and important contribution to its field?

Acceptable

General interest: Is the paper of sufficient general interest?

Acceptable

Quality of the paper: Is the overall quality of the paper suitable?

Acceptable

Is the length of the paper justified?

Yes

Should the paper be seen by a specialist statistical reviewer?

No

Do you have any concerns about statistical analyses in this paper? If so, please specify them explicitly in your report.

Yes

It is a condition of publication that authors make their supporting data, code and materials available - either as supplementary material or hosted in an external repository. Please rate, if applicable, the supporting data on the following criteria.

Is it accessible?

Yes

Is it clear?

Yes

Is it adequate?

Yes

Do you have any ethical concerns with this paper?

No

Comments to the Author

See attached document. (See Appendix A)

Decision letter (RSPB-2020-3213.R0)

24-Feb-2021

Dear Mr Travers:

I am writing to inform you that your manuscript RSPB-2020-3213 entitled "Habitat patches providing South-North connectivity are under-protected in a fragmented landscape" has, in its current form, been rejected for publication in Proceedings B.

This action has been taken on the advice of referees, who have recommended that substantial revisions are necessary. With this in mind we would be happy to consider a resubmission, provided the comments of the referees are fully addressed. However please note that this is not a provisional acceptance.

Sincerely,
Dr Maurine Neiman
[mailto: proceedingsb@royalsociety.org](mailto:proceedingsb@royalsociety.org)

Associate Editor
Comments to Author:

Two referees provide coherent and comprehensive comments on this manuscript assessing the capacity of protected area networks to facilitate range shifts. The referees assess that substantial work is needed to justify the importance and novelty of the research in relation to climate change adaptation in conservation. They request greater explanation, clarification and contextualisation of the approaches taken and results obtained, especially in terms of both 1) the advantages and advances of the Condatis circuit theory technique relative to alternatives, and 2) the realism of the

approach using individual habitat types, its applicability to particular species, and the rationale in terms of south-north range shifts in the landscapes studied.

Reviewer(s)' Comments to Author:

Referee: 1

Comments to the Author(s)

This is an interesting study, using an adaptation of circuit theory to add nuance to connectivity analyses that are relevant for climate change impact analyses. In general, I appreciated the addition of more biology to the circuit approach, which has several limitations for biological interpretation. I offer the following remarks to strengthen this contribution.

I found the logic supporting the methods and model design choices to be insufficient. The methodological choices don't seem to match the research questions. More logic and explanation is required to make the leap to experimental design. Right now, it seems the approach was designed and tested, then a question was built around the results.

Habitat and patches are not defined. The species that are relevant to this analysis are not mentioned until the conclusions. It is unclear why the authors chose their habitat dataset, nor why they split it up into so many habitat types. If the intent is a broad-scale, multi-species perspective on PAs, I don't see how this approach really does this. I'd expect to see an analysis that runs on a map that integrates all land-cover types. If highlighting specific land-cover types is a key objective, that should be made explicit and explained.

Introduction

38 – synergistic seems an overstatement in a general statement. Additive or coinciding may be true, impacts are not always interacting, multiplicative. Adding qualifying language like “Can” is likely more general and appropriate.

44-48 – repetitive with respect to the multi-generational statements. Condense.

50. Whilst – replace with synonym as meaning is unclear.

72- “as within the Maya.. to Berryessa project in CA”. Suggest you omit or elaborate to describe this project and how it demonstrates your point. There is too little information in the excerpt to be meaningful.

76-77 – For what kind of species have S-N shifts have been documented? This is a good place to indicate what kind of species your analysis will represent (or be relevant for) – plants, terrestrial animal, mammals, birds?

78 – How is habitat defined, if a species-specific view is not used? Suggest omitting ‘habitat’ and replacing it with a more general term.

82 – How is a patch defined?

Methods

93-116. Move after methods/Conclatis overview (or integrate within).

153 – unclear.

Results

170. It is still unclear to my why you chose this dataset as representative of habitat. What about species that use more one type of landcover? Why break this apart into types? This doesn't seem needed based on your stated research questions.

196 – Replace ‘approaches’ with a synonym.

227 – omit: ‘we offer a cost-effective remedy’. This is overstated.

228-239: The reads more like introduction than discussion.

254 – if Aichi target is now defunct, why mention it in the introduction? Perhaps substitute another example.

Referee: 2

Comments to the Author(s)

See attached document.

Author's Response to Decision Letter for (RSPB-2020-3213.R0)

See Appendix B.

RSPB-2021-1010.R0

Review form: Reviewer 1

Recommendation

Major revision is needed (please make suggestions in comments)

Scientific importance: Is the manuscript an original and important contribution to its field?

Good

General interest: Is the paper of sufficient general interest?

Good

Quality of the paper: Is the overall quality of the paper suitable?

Acceptable

Is the length of the paper justified?

Yes

Should the paper be seen by a specialist statistical reviewer?

No

Do you have any concerns about statistical analyses in this paper? If so, please specify them explicitly in your report.

No

It is a condition of publication that authors make their supporting data, code and materials available - either as supplementary material or hosted in an external repository. Please rate, if applicable, the supporting data on the following criteria.

Is it accessible?

Yes

Is it clear?

Yes

Is it adequate?

Yes

Do you have any ethical concerns with this paper?

No

Comments to the Author

This is an improved version of the manuscript, with more detail and explanation.

My main comment is that I still don't think the authors adequately explained why the analysis is conducted for different habitat types, and why it makes sense to interpret large-scale connectivity in this way. The reader needs to understand not only the practical reason why England

designates land this way (which the authors now partially explain), but more importantly why the analysis makes sense to conduct and interpret in this way. I appreciate that this is noted in the caveats section but I still wonder how useful the results are if they are not combined in some way to express overall connectivity of all patches and PAs – from both biological and planning perspectives.

Needs a strong proofread for grammar, punctuation, and missing words.

23-24 – Awkward phrasing. Suggest: in fragmented landscapes, some patches are more important than others in maintaining population or habitat or landscape connectivity... or similar. I would omit the assertion that methods for identifying key patches are not widely used as it is difficult to assess the accuracy of this vague statement.

25 – Few know what the Condatis methodology is. I suggest adding a brief statement that alludes to the type of modeling approach.

28 – Remove ‘concerning.’ This is a value judgement. Replace with a number or objective description.

30 – consider changing ‘likely’ to ‘may be’ as you did not assess any other countries or the probability of having a similar situation, or reference as in 257.

34 – Remove ‘shows promising results’ as you don’t really evaluate whether this approach does better than other connectivity-based approaches. Suggest the message be re-framed as consideration of connectivity (using this method) provides an efficient means of identifying additional protected areas that prioritizes connectivity.

54 – I don’t see this as a recent debate, just the re-emergence of the debate. Suggest removing ‘recent’ and briefly explaining what the new argument is.

55 – Need more words, citations, and qualifiers. Aggregated patches are better for many but not all species. What kind of habitat creation models are you referring to?

64-65: Please add citations or state more speculatively.

67 - This citation for recently putting connectivity science into practice is now a decade old, and there are now several examples of this being done. Suggest updating the tone message of this passage to suggest that there is a current movement to model and consider climate connectivity, but there is need for additional tools and methods to [insert a statement that defines your unique contribution].

72 – It would be helpful to add a statement that explicitly states why your analysis is divided out into different habitat types (and hence why your analysis is structured as such). What does ecologically distinct really mean in the context of your analysis? Does each type describe a unique subset of species with unique habitat requirements, that do not use other ecotypes? You have provided new content and explanations here and elsewhere, but I still don’t understand why the analysis is conducted separately for each type.

78 – N-S shifts: Are these documented for your study area? If so, state this or where this comment applies to.

88 – Is ‘speed’ the right word to use here? Seem in contradiction to circuit-theory interpretation that is more adept at describing movement resistance and landscape patterns, not speed.

109 – I don’t think circuit-based interpretations are limited to single movements by individuals. It represents movement as current through a landscape, with little consideration of actual movement distances. It doesn’t always model steppingstone movement i.e., one patch to another (unless the analysis is explicitly set up to do so) – which is maybe more of what you mean to communicate? Please clarify.

120 – Please highlight that this analysis doesn’t consider matrix heterogeneity that may affect accessibility of patches and functional connectivity of the landscape. This then prompts the question of how would your results changed if you considered matrix heterogeneity including barriers to movement? Accessibility is a big part of connectivity. Without this, the analysis seems to rely on distance and required movement from S to N, regardless of topography and other movement modifiers.

166 – Did you conduct a sensitivity analysis on your choice of 100 as R? How did you decide this was an appropriate value?

262 - 263. If you are forcing current S to N without regard for matrix heterogeneity, I don't think you are really accounting for 'the important routes ... species are likely to take'. Suggest making this a more accurate statement. They are potential regions in which many species may move if tracking S-N shifts in the absence of movement constraints (e.g., topography), barriers etc.

Review form: Reviewer 2

Recommendation

Accept with minor revision (please list in comments)

Scientific importance: Is the manuscript an original and important contribution to its field?

Acceptable

General interest: Is the paper of sufficient general interest?

Good

Quality of the paper: Is the overall quality of the paper suitable?

Acceptable

Is the length of the paper justified?

Yes

Should the paper be seen by a specialist statistical reviewer?

No

Do you have any concerns about statistical analyses in this paper? If so, please specify them explicitly in your report.

No

It is a condition of publication that authors make their supporting data, code and materials available - either as supplementary material or hosted in an external repository. Please rate, if applicable, the supporting data on the following criteria.

Is it accessible?

Yes

Is it clear?

Yes

Is it adequate?

Yes

Do you have any ethical concerns with this paper?

No

Comments to the Author

Review of: Habitat patches providing South-North connectivity are under-protected in a fragmented landscape

Summary: The authors' study is aimed at understanding north-south habitat connectivity, and the degree that certain habitat types are protected and contribute to connectivity, to inform the prioritization of area-based conservation that supports climate connectivity.

The authors have done a really great job of making the requested changes to the manuscript. However, they need to go further in their explanation of the Condatis method. They have stated that they preferred to reference literature rather than bog readers down in the details, and I agree, but they need a better balance because these methods are so uncommon and do not match perfectly to typical uses of circuit theory. Many readers will be coming with that typical use background. I outlined the biggest of these question marks in my detailed comments below. The other big hole is the assumption that the matrix doesn't affect movement and that there is no variability in matrix effects on movement or breeding. I think there are three options here: (1) incorporate some piece of the matrix (e.g., road), (2) frame the study around structural connectivity (I think stepping stones can fit into this framing) and/or make it more clear what the study can and cannot do, and (3) hone in on several case study areas where roads and urban areas could block movement, as a description of the studies limitations or, better yet, next steps. These are just some ideas, but I think #2 is the best given the stage of the study. Is it possible to include any information about the matrix with this methodology? If so, this could be a future step. If not, this could be mentioned as a limitation.

Response to previous comments (Reviewer 2):

Comment regarding contiguous habitat: Though not the point of the comment, I agree it is definitely not the case that species need contiguous habitat to move across landscapes. The authors state this, then say that many species are shifting their ranges through fragmented, non-contiguous habitat. However, the Condatis method they have described does not account for the fragmented landscape outside of habitat cells of interest. Because of that, using circuit theory, areas of contiguous habitat will jump out as high flow. If the matrix were included, it is possible that some of these areas, stepping stones in particular, are actually cut off by roads of urban areas.

Comment about Condatis needing more explanation: See detailed comments below about the need for more information to make this methodology clearer. Removal of a resistance surface in particular needs explanation, as a circuit is needed in circuit theory, and a circuit contains resistors.

Detailed comments:

1. Line 85: Strict protection isn't the only conservation intervention available for promoting/safeguarding connectivity, and often it is the less feasible option for conserving smaller patches in human-dominated landscapes.
2. Line 133: This could be "targeted conservation" rather than protection to be more flexible than relying strictly on the addition of protected area.
3. Line 143: Replace like with likely. Is there a reference for this sentence?
4. Line: 145-147: Remove this sentence or explain it further either here or below. In particular, "consideration of colonization events between non-neighboring cells".
5. Paragraph at 148: This needs more information: How is resistance coded? Even though you say the need for a traditional resistance surface is removed, there certainly still is a resistance surface for which current can flow? Why are you assuming the matrix is homogenous, when many areas are impermeable to movement? Source nodes are located in the south; are those habitat cells? An equation to define how the dispersal kernel is parameterized is also needed as well as an explanation of how this affects the analysis in circuit theory terms? (I see some of this in the Appendix, but that is hard to follow) If the source nodes are located in the south and there is a dispersal distance that limits the size of the circuit, then how can the ground nodes be located in the north beyond this distance? I'm assuming there's some iteration (the time-steps mentioned below?) but it is hard to follow.
6. Line 160: Remove "Given this definition".
7. Line 161: Network of habitat patches or protected areas? How is conductance calculated? Is this total conductance across the circuit?
8. Line 162: Importance of each habitat cell? What kind of cell? What is flow - current density?
9. Line 165: Replace "countless" with "all" possible travel routes.

10. Line 166: Follow the sentence starting with “Flow” with an explanation of what that means in practical terms – e.g., the degree to which connectivity would decline if that cell of habitat was lost.
11. Lines 218-220: Are these values used as conductance (or the inverse as resistance)?
12. Section at line 221: There are some explanations of things in here that should follow directly below/within the section Condatis analysis. Can you combine? It is not clear how reproductive rate comes into the analysis. I see it in your dispersal kernel in the Appendix, but it needs some explanation in the main text, and this may be better suited in the second paragraph of Condatis Analysis.
13. Line 225: There’s some wording errors here.
14. Line 252-253: Move what you have in parentheses to after the previous sentence mentioning R fixed at 100, so we know what that is supposed to mean.
15. Line 263-267: This is hard to follow with the wording, suggest removing “so we assigned flow from cells to patches”. The supplemental figure is really key to understanding this- great figure!
16. Line 266: change 1km/2km to 1km or 2km.
17. Line 308: What does that mean, they were too few, sparsely distributed and fine in scale?
18. Line 342: Did you try a model with just flow?
19. Line 376: remove “exist to”
20. Line 414: Conductance gains less than the increase in area. Do you mean proportional increases in conductance and area?
21. Line 419: This study does not examine landscape permeability because it ignores the matrix. So maybe it’s better to say safeguarding connected habitats and patches that act as step stones. The end of this sentence should be changed to connectivity rather than conductance; conservation planners are not discussing conductance usually.
22. Line 420-423: The beginning of this sentence needs some rewording.
23. Paragraph at 424: “biased towards low-flow patches despite also being biased towards large patches with high flow” is contradictory and needs rewording and/or explanation. The last sentence in this paragraph is also confusing.
24. Paragraph at 452: I think it’s a good idea to also mention that you didn’t examine the matrix. Some high flow areas may not be high flow at all if separated by massive urban areas and roads.
25. Fig 2. Do you mean “proportion of protected total flow” here in (b)?
26. Fig. 5 This is a hard to understand title. In (a), you say current conductance, but the y axis says total flow?

Decision letter (RSPB-2021-1010.R0)

24-May-2021

Dear Mr Travers:

Your manuscript has now been peer reviewed and the reviews have been assessed by an Associate Editor. The reviewers’ comments (not including confidential comments to the Editor) and the comments from the Associate Editor are included at the end of this email for your reference. As you will see, the reviewers and the Editors have raised some concerns with your manuscript and we would like to invite you to revise your manuscript to address them.

Research ethics:

Use of animals and field studies:

It is a condition of publication that you make available the data and research materials supporting the results in the article (<https://royalsociety.org/journals/authors/author-guidelines/#data>). Datasets should be deposited in an appropriate publicly available repository and details of the associated accession number, link or DOI to the datasets must be included in the Data Accessibility section of the article (<https://royalsociety.org/journals/ethics-policies/data-sharing-mining/>). Reference(s) to datasets should also be included in the reference list of the article with DOIs (where available).

Please submit a copy of your revised paper within three weeks. If we do not hear from you within this time your manuscript will be rejected. If you are unable to meet this deadline please let us know as soon as possible, as we may be able to grant a short extension.

Best wishes,
Dr Maurine Neiman
mailto:proceedingsb@royalsociety.org

Associate Editor Board Member

Comments to Author:

Both reviewers of the resubmitted manuscript were grateful to the authors for the changes made, and noted that the resubmission was improved over the original submission. The reviewers do however provide a clear set of questions and recommendations regarding remaining areas that would benefit from greater clarification and contextualisation - particularly related to the approach and its implications. The reviewers also provide a number of specific comments about elements of the text.

Reviewer(s)' Comments to Author:

Referee: 2

Comments to the Author(s).

Review of: Habitat patches providing South-North connectivity are under-protected in a fragmented landscape

Summary: The authors' study is aimed at understanding north-south habitat connectivity, and the degree that certain habitat types are protected and contribute to connectivity, to inform the prioritization of area-based conservation that supports climate connectivity.

The authors have done a really great job of making the requested changes to the manuscript. However, they need to go further in their explanation of the Condatis method. They have stated that they preferred to reference literature rather than bog readers down in the details, and I agree, but they need a better balance because these methods are so uncommon and do not match perfectly to typical uses of circuit theory. Many readers will be coming with that typical use background. I outlined the biggest of these question marks in my detailed comments below. The other big hole is the assumption that the matrix doesn't affect movement and that there is no variability in matrix effects on movement or breeding. I think there are three options here: (1) incorporate some piece of the matrix (e.g., road), (2) frame the study around structural connectivity (I think stepping stones can fit into this framing) and/or make it more clear what the study can and cannot do, and (3) hone in on several case study areas where roads and urban areas could block movement, as a description of the studies limitations or, better yet, next steps. These are just some ideas, but I think #2 is the best given the stage of the study. Is it possible to include any information about the matrix with this methodology? If so, this could be a future step. If not, this could be mentioned as a limitation.

Response to previous comments (Reviewer 2):

Comment regarding contiguous habitat: Though not the point of the comment, I agree it is definitely not the case that species need contiguous habitat to move across landscapes. The authors state this, then say that many species are shifting their ranges through fragmented, non-contiguous habitat. However, the Condatis method they have described does not account for the fragmented landscape outside of habitat cells of interest. Because of that, using circuit theory, areas of contiguous habitat will jump out as high flow. If the matrix were included, it is possible that some of these areas, stepping stones in particular, are actually cut off by roads or urban areas.

Comment about Condatis needing more explanation: See detailed comments below about the need for more information to make this methodology clearer. Removal of a resistance surface in particular needs explanation, as a circuit is needed in circuit theory, and a circuit contains resistors.

Detailed comments:

1. Line 85: Strict protection isn't the only conservation intervention available for promoting/safeguarding connectivity, and often it is the less feasible option for conserving smaller patches in human-dominated landscapes.
2. Line 133: This could be "targeted conservation" rather than protection to be more flexible than relying strictly on the addition of protected area.
3. Line 143: Replace like with likely. Is there a reference for this sentence?
4. Line: 145-147: Remove this sentence or explain it further either here or below. In particular, "consideration of colonization events between non-neighboring cells".
5. Paragraph at 148: This needs more information: How is resistance coded? Even though you say the need for a traditional resistance surface is removed, there certainly still is a resistance surface for which current can flow? Why are you assuming the matrix is homogenous, when many areas are impermeable to movement? Source nodes are located in the south; are those habitat cells? An equation to define how the dispersal kernel is parameterized is also needed as well as an explanation of how this affects the analysis in circuit theory terms? (I see some of this in the Appendix, but that is hard to follow) If the source nodes are located in the south and there is a dispersal distance that limits the size of the circuit, then how can the ground nodes be located in the north beyond this distance? I'm assuming there's some iteration (the time-steps mentioned below?) but it is hard to follow.
6. Line 160: Remove "Given this definition".
7. Line 161: Network of habitat patches or protected areas? How is conductance calculated? Is this total conductance across the circuit?
8. Line 162: Importance of each habitat cell? What kind of cell? What is flow - current density?
9. Line 165: Replace "countless" with "all" possible travel routes.
10. Line 166: Follow the sentence starting with "Flow" with an explanation of what that means in practical terms - e.g., the degree to which connectivity would decline if that cell of habitat was lost.
11. Lines 218-220: Are these values used as conductance (or the inverse as resistance)?
12. Section at line 221: There are some explanations of things in here that should follow directly below/within the section Condatis analysis. Can you combine? It is not clear how reproductive rate comes into the analysis. I see it in your dispersal kernel in the Appendix, but it needs some explanation in the main text, and this may be better suited in the second paragraph of Condatis Analysis.
13. Line 225: There's some wording errors here.
14. Line 252-253: Move what you have in parentheses to after the previous sentence mentioning R fixed at 100, so we know what that is supposed to mean.
15. Line 263-267: This is hard to follow with the wording, suggest removing "so we assigned flow from cells to patches". The supplemental figure is really key to understanding this- great figure!
16. Line 266: change 1km/2km to 1km or 2km.
17. Line 308: What does that mean, they were too few, sparsely distributed and fine in scale?
18. Line 342: Did you try a model with just flow?

19. Line 376: remove “exist to”
20. Line 414: Conductance gains less than the increase in area. Do you mean proportional increases in conductance and area?
21. Line 419: This study does not examine landscape permeability because it ignores the matrix. So maybe it's better to say safeguarding connected habitats and patches that act as step stones. The end of this sentence should be changed to connectivity rather than conductance; conservation planners are not discussing conductance usually.
22. Line 420-423: The beginning of this sentence needs some rewording.
23. Paragraph at 424: “biased towards low-flow patches despite also being biased towards large patches with high flow” is contradictory and needs rewording and/or explanation. The last sentence in this paragraph is also confusing.
24. Paragraph at 452: I think it's a good idea to also mention that you didn't examine the matrix. Some high flow areas may not be high flow at all if separated by massive urban areas and roads.
25. Fig 2. Do you mean “proportion of protected total flow” here in (b)?
26. Fig. 5 This is a hard to understand title. In (a), you say current conductance, but the y axis says total flow?

Referee: 1

Comments to the Author(s).

This is an improved version of the manuscript, with more detail and explanation.

My main comment is that I still don't think the authors adequately explained why the analysis is conducted for different habitat types, and why it makes sense to interpret large-scale connectivity in this way. The reader needs to understand not only the practical reason why England designates land this way (which the authors now partially explain), but more importantly why the analysis makes sense to conduct and interpret in this way. I appreciate that this is noted in the caveats section but I still wonder how useful the results are if they are not combined in some way to express overall connectivity of all patches and PAs – from both biological and planning perspectives.

Needs a strong proofread for grammar, punctuation, and missing words.

23-24 – Awkward phrasing. Suggest: in fragmented landscapes, some patches are more important than others in maintaining population or habitat or landscape connectivity... or similar. I would omit the assertion that methods for identifying key patches are not widely used as it is difficult to assess the accuracy of this vague statement.

25 – Few know what the Condatis methodology is. I suggest adding a brief statement that alludes to the type of modeling approach.

28 – Remove ‘concerning.’ This is a value judgement. Replace with a number or objective description.

30 – consider changing ‘likely’ to ‘may be’ as you did not assess any other countries or the probability of having a similar situation, or reference as in 257.

34 – Remove ‘shows promising results’ as you don't really evaluate whether this approach does better than other connectivity-based approaches. Suggest the message be re-framed as consideration of connectivity (using this method) provides an efficient means of identifying additional protected areas that prioritizes connectivity.

54 – I don't see this as a recent debate, just the re-emergence of the debate. Suggest removing ‘recent’ and briefly explaining what the new argument is.

55 – Need more words, citations, and qualifiers. Aggregated patches are better for many but not all species. What kind of habitat creation models are you referring to?

64-65: Please add citations or state more speculatively.

67 - This citation for recently putting connectivity science into practice is now a decade old, and there are now several examples of this being done. Suggest updating the tone message of this passage to suggest that there is a current movement to model and consider climate connectivity,

but there is need for additional tools and methods to [insert a statement that defines your unique contribution].

72 - It would be helpful to add a statement that explicitly states why your analysis is divided out into different habitat types (and hence why your analysis is structured as such). What does ecologically distinct really mean in the context of your analysis? Does each type describe a unique subset of species with unique habitat requirements, that do not use other ecotypes? You have provided new content and explanations here and elsewhere, but I still don't understand why the analysis is conducted separately for each type.

78 - N-S shifts: Are these documented for your study area? If so, state this or where this comment applies to.

88 - Is 'speed' the right word to use here? Seem in contradiction to circuit-theory interpretation that is more adept at describing movement resistance and landscape patterns, not speed.

109 - I don't think circuit-based interpretations are limited to single movements by individuals. It represents movement as current through a landscape, with little consideration of actual movement distances. It doesn't always model steppingstone movement i.e., one patch to another (unless the analysis is explicitly set up to do so) - which is maybe more of what you mean to communicate? Please clarify.

120 - Please highlight that this analysis doesn't consider matrix heterogeneity that may affect accessibility of patches and functional connectivity of the landscape. This then prompts the question of how would your results changed if you considered matrix heterogeneity including barriers to movement? Accessibility is a big part of connectivity. Without this, the analysis seems to rely on distance and required movement from S to N, regardless of topography and other movement modifiers.

166 - Did you conduct a sensitivity analysis on your choice of 100 as R? How did you decide this was an appropriate value?

262 - 263. If you are forcing current S to N without regard for matrix heterogeneity, I don't think you are really accounting for 'the important routes ...species are likely to take'. Suggest making this a more accurate statement. They are potential regions in which many species may move if tracking S-N shifts in the absence of movement constraints (e.g., topography), barriers etc.

Author's Response to Decision Letter for (RSPB-2021-1010.R0)

See Appendix C.

RSPB-2021-1010.R1 (Revision)

Review form: Reviewer 1

Recommendation

Accept with minor revision (please list in comments)

Scientific importance: Is the manuscript an original and important contribution to its field?

Good

General interest: Is the paper of sufficient general interest?

Good

Quality of the paper: Is the overall quality of the paper suitable?

Good

Is the length of the paper justified?

Yes

Should the paper be seen by a specialist statistical reviewer?

No

Do you have any concerns about statistical analyses in this paper? If so, please specify them explicitly in your report.

No

It is a condition of publication that authors make their supporting data, code and materials available - either as supplementary material or hosted in an external repository. Please rate, if applicable, the supporting data on the following criteria.

Is it accessible?

Yes

Is it clear?

Yes

Is it adequate?

Yes

Do you have any ethical concerns with this paper?

No

Comments to the Author

Thank you for addressing my previous comments and suggestions. I find this draft to be much improved. I have a few minor suggestions for further improvement. 1) Please edit to improve sentence structure and phrasing (e.g., 68). 2) Please re-consider the order of your discussion paragraphs or work on the transitions between ideas. I found this section a bit disjointed.

Specific Comments

38-40. Rephrase.

45. omit 'furthermore'

66. replace 'probably' with 'generally'

75. 'form platforms'?

81. omit 'meanwhile during recent climatic warming', replace with 'Contemporary climatic warming'...

85. rephrase.

Methods are written in a mix of tenses.

153. Clearly re-state here that unlike circuit-based analysis, a resistance surface/calculation only represents dispersal distances, and not the expected difficulty of moving through the landscape (as in circuitscape or omniscapes). Because this information is excluded, movement barriers are not included.

163-170. Emphasize how the interpretation is different than simple circuit or least cost analyses. I.e., Connectivity from a multi-generational dispersal perspective; not landscape permeability due to movement ease/difficulty (without movement constraints as in traditional circuit analyses), or shortest path between 2 patches etc..

174 - what do you mean by traits and processes?

270 - I'm not sure if you are talking about patches within a PA network or smaller patches within a larger patch. Rephrase.

Decision letter (RSPB-2021-1010.R1)

23-Jul-2021

Dear Mr Travers

I am pleased to inform you that your manuscript RSPB-2021-1010.R1 entitled "Habitat patches providing South-North connectivity are under-protected in a fragmented landscape" has been accepted for publication in Proceedings B.

The referee(s) have recommended publication, but also suggest some minor revisions to your manuscript. Therefore, I invite you to respond to the referee(s)' comments and revise your manuscript. Because the schedule for publication is very tight, it is a condition of publication that you submit the revised version of your manuscript within 7 days. If you do not think you will be able to meet this date please let us know.

Sincerely,

Dr Maurine Neiman

Associate Editor:

Comments to Author:

The reviewer was pleased with the revisions made to the manuscript, but recommends that the authors make a few remaining improvements to presentation.

Reviewer(s)' Comments to Author:

Referee: 1

Comments to the Author(s)

Thank you for addressing my previous comments and suggestions. I find this draft to be much improved. I have a few minor suggestions for further improvement. 1) Please edit to improve sentence structure and phrasing (e.g., 68). 2) Please re-consider the order of your discussion paragraphs or work on the transitions between ideas. I found this section a bit disjointed.

Specific Comments

38-40. Rephrase.

45. omit 'furthermore'

66. replace 'probably' with 'generally'

75. 'form platforms'?

81. omit 'meanwhile during recent climatic warming', replace with 'Contemporary climatic warming'...

85. rephrase.

Methods are written in a mix of tenses.

153. Clearly re-state here that unlike circuit-based analysis, a resistance surface/calculation only represents dispersal distances, and not the expected difficulty of moving through the landscape (as in circuitscape or omniscap). Because this information is excluded, movement barriers are not included.

163-170. Emphasize how the interpretation is different than simple circuit or least cost analyses.

I.e., Connectivity from a multi-generational dispersal perspective; not landscape permeability due to movement ease/difficulty (without movement constraints as in traditional circuit analyses), or shortest path between 2 patches etc..

174 - what do you mean by traits and processes?

270 - I'm not sure if you are talking about patches within a PA network or smaller patches within a larger patch. Rephrase.

Author's Response to Decision Letter for (RSPB-2021-1010.R1)

See Appendix D.

Decision letter (RSPB-2021-1010.R2)

02-Aug-2021

Dear Mr Travers

I am pleased to inform you that your manuscript entitled "Habitat patches providing South-North connectivity are under-protected in a fragmented landscape" has been accepted for publication in Proceedings B.

Data Accessibility section

Open Access

You are invited to opt for Open Access, making your freely available to all as soon as it is ready for publication under a CCBY licence. Our article processing charge for Open Access is £1700. Corresponding authors from member institutions

Paper charges

Sincerely,

Appendix A

Review: Habitat patches providing south-north connectivity are under-protected in a fragmented landscape.

Summary: The authors' study is aimed at understanding the north-south contiguity of a variety of natural habitat types, and the degree that these habitat types are fragmented and protected, to inform the prioritization of area-based conservation that supports climate connectivity.

Major comments: I think this is an interesting idea and approach but as it stands there are some missing pieces. It's not clear how north-south contiguity of habitat types by themselves will provide species with the ability to adapt to climate change, particularly when species are likely to move through multiple habitat types during the process of range expansion. This seems to be an analysis of the contiguity of habitat types, and thus akin to structural connectivity, but the authors haven't convinced me that this approach will be the best way to prioritize future protected area designation, as it lacks that linkage to species of conservation concern and/or climate-related range expansions across the region. And while I'm intrigued by the Condatis method, it's not clear why a circuit theory approach is best rather than just identifying contiguous habitat. Also, Condatis is an uncommon approach and as such needs more explanation than what is provided for the reader to understand the theory, the output and how these differ from more common approaches like Circuitscape and least-cost paths.

Line by line comments:

1. Line 40: I would just delete the sentence starting with "Local management..." because it feels a little out of place and then allows you to connect your sentence about connectivity in conservation planning to how people study/or don't study connectivity.
2. Line 46: "Demographic momentum for multi-generational range shifts" is not referenced and it's not clear how this would happen. Do you mean the population will grow to a point where the likelihood of emigration increases?
3. Line 46-49: So, considering multi-generational connectivity is important because the effects of climate change will be slow or because species responses will be slow? Just more clarity here to understand the issue would be great.
4. Line 67: The previous sentence implies that existing PA networks are enough to cover projected range expansion. If that is the case, then why would additional patches be critical for range expansion? It's also not clear what 'projection of patches' means in this sentence. Do you mean projected range for expansion?
5. Line 70: Can this be changed to "However there has been a significant delay in putting connectivity science into practice..." or rewrite that first part of the sentence to be more specific about what evidence and how long the lag can be?
6. Line 72-73: The example at the end of the sentence comes without context or explanation. Can you provide more detail here so the reader can see how this fits?
7. Line 78: Explain what "high-flow" means here.
8. Line 82: The difference between circuit theory in Condatis and other uses of circuit theory isn't quite clear. This is something that a lot of people will be interested in so I recommend you take another sentence or two to clarify. Is it just a conceptual difference – dealing with colonists versus dispersers? How is circuit theory used differently (maybe put in methods but more clarity would great).

9. Last paragraph in Introduction: It's not clear what species or taxonomic group you're interested in here, which seems like a vital consideration given you're interested in long-distance, multi-generational connectivity for colonists and because the ability to track a changing climate also depends on species habitat requirements. If you're going for a species-agnostic approach it's important you state that with a brief explanation why and how doing that can still answer your question/inform conservation.
10. Lines 99-100: I would replace shapefile format with vector format instead.
11. Lines 102-103: Can you be more specific about what "works best" means here? Perhaps something about the trade-off between resolution and memory. This could go into the supplemental info, but either way I think the readers would be interested in this.
12. Lines 105-106: By "accurate measure" do you mean continuous value rather than categorical? I think it would be better to use those terms rather than accurate measure, unless you've gone out and ground-truthed the amount of forest cover and confirmed some level of accuracy.
13. Line 113: Delete: "especially in the context of climate change."
14. Line 116: Can you present the number of patches lost due to the rasterization – maybe with the proportion of area that these patches represent compared to other patches. That would really drive home that removal of these patches will have little consequence.
15. Line 121: Can you elaborate on "... as if the network were an electrical circuit." I think you need a bit more detail before we can see how it's like an electrical circuit. (e.g., grid cells connected by resistors etc)
16. Lines 122-123: Can you elaborate on how both dispersal rate and reproductive rate can be used to parameterize resistance? Does a cell depict the value of one of those or both in combination? How do you obtain these values? Are they empirically estimated? And did you conduct any sensitivity analysis to these values?
17. Line 128: Can you clarify what you mean by this sentence? So, high connectivity value in a cell means high flow and high change in flow if that cell was removed from the landscape? Maybe you just need to clarify what "focal cell" is referring to?
18. Lines 138-139: Can you provide some explanation about why you chose 100 as the reproductive rate? Is this high or low, is it representative of a particular expectation, are you testing multiple values and then validating with observational data? Something similar to how you described the dispersal rate choice would be a great start. How are you varying reproductive rate by habitat type? I think you're not, but it's not clear then how resistance varies across the landscape. I'm wondering if you need a diagram to explain this...
19. Line 143: What is meant by the largest gap in the network? This makes me wonder if you are examining the connectivity only of particular habitats, separately – so this is a structural connectivity analysis where you code grid cells of a particular habitat as having conductance and all other grid cells have no conductance? If this is the case, I think it would be helpful to understand how this could inform conservation, particularly when species are likely to move across different habitat types. Essentially, animals won't necessarily see a 'gap' in the network when moving from one patch to another when expanding their range. I guess I'm just not convinced that range expansion has to occur along a contiguous habitat type.
20. Line 148: What is a Moore neighborhood?

21. Paragraph starting at line 145: So really you are getting the fraction of flow for habitat patches that are smaller than the 1km²? Another diagram would be great here perhaps if that's easier than explaining in the text, but that diagram could go in the supplemental info.
22. Line 159: Do you mean flow when you say conductance? Or are you looking at the inverse of the resistance surface here? If it's a different metric maybe you could repeat what it is here, and how it may differ from the typical definition of conductance in circuit theoretic approaches to connectivity modeling.
23. Line 160: Have you defined what a habitat network is yet? Does that mean all patches of a particular habitat? I think you first mentioned this in the intro, but you need to make it clear what you mean by that given you have been mentioning patches and habitats separately as well. Maybe in this instance you just mean habitat type. This also comes up again in line 164 and I'm wondering if you mean habitat type here as well.
24. Line 165: So low values signify reduced interior habitat, but reduced compared to what? Is it reduced compared to a completely contiguous patch?
25. Lines 166-168: It is not clear what you're doing here. I think we need a lot more detail.
26. Line 170: Again, I feel like habitat network is strange here. It's a habitat type and you're just describing the area of that habitat type, but maybe is a network of patches of a particular habitat type and you're describing the area of a contiguous network of patches? I'm sure its just semantics, but right now I'm a bit confused.
27. Other comments on methods: I think you should pull in some of the detail from your supp info regarding Condatis and circuit theory and description of conductance in this method to help clarify what you're doing.
28. Lines 175-183: This doesn't seem to be very surprising that more contiguous habitat have higher conductance and that larger patches are protected and more fragmentation occurs outside formal protection. Is there something more to it? Please do explain if there is!
29. Line 191: Is blanket bog interesting enough to pull out here in the text? Is this an important habitat for threatened species or climate change? If not you don't need to call it out.
30. Lines 195-202: I'm wondering if it's appropriate to include both flow and area as predictors if they are correlated themselves and you are trying to interpret their coefficient value.
31. Lines 203-213: This is interesting in that you don't have to protect a ton of area to get huge gains in flow protection (which is really structural connectivity). Going from 1-10% area has a much greater gain than going from 1—25%... But then it looks like this is contradicted in the last couple of sentences where area and flow are again tightly correlated (which makes sense!). Can this be explained more?
32. Other comments on results: There were no figure legends unfortunately, so it was difficult to understand your findings. I think you should move the supp info fig 1 into the main figures with maybe some habitat information and a protected area layer as this will help clarify the methods and goals.
33. Lines 216-218: Can you provide any explanation why? Are these areas also valuable for human development whereas protected areas tend to be areas of less value to humans? I see you get to this later on, but maybe you should elevate this discussion to this first paragraph.
34. Line 222-223: While I agree with you in part, it is also because these habitats are undisturbed, which is a result of protection.

35. Line 224-226: I just want to point out that focusing solely on structural connectivity as you do in this study, rather than functional connectivity, and ignoring the connectedness of different habitat types in the role of range expansion is not exactly ecological realism either.
36. Lines 241-243: You show that contiguity of habitat type is lacking in many cases in the north-south direction and that many of these areas are not protected, but we don't actually see that these patches are important for range expansion or adaptation to climate change. Could you try and validate these models in some way using movement data or species distributions over time?

Appendix B

Comment	Response	Line(s) in tracked changes document	Edit Code in tracked changes document
Referee: 1			
This is an interesting study, using an adaptation of circuit theory to add nuance to connectivity analyses that are relevant for climate change impact analyses. In general, I appreciated the addition of more biology to the circuit approach, which has several limitations for biological interpretation. I offer the following remarks to strengthen this contribution.	We thank you for your time and interest in this study. We are pleased you appreciate our novel approach, in which circuit theory provides a computationally-efficient analogy for metapopulation processes. We are very grateful for your comments, which are both clear and constructive. We have made significant changes as detailed below.		
I found the logic supporting the methods and model design choices to be insufficient. The methodological choices don't seem to match the research questions. More logic and explanation is required to make the leap to experimental design. Right now, it seems the approach was designed and tested, then a question was built around the results.	We have now added detail to our introduction and methods that we hope better justify our choices, and explain the rationale of our study design. Notably a new paragraph in the introduction explaining the legislative basis for our approach, and a re-written methods section which we feel better explains the Condatis methodology.	83–93, 121–148	I.8, M.2
Habitat and patches are not defined.	We have now added detail to the penultimate paragraph of the introduction to address this.	96-99, 101-102	I.9, I.11
The species that are relevant to this analysis are not mentioned until the conclusions.	We apologise for this omission. We now specifically reference the broad suite of legally protected species we hoped to represent in this study. We also highlight that the focal priority habitats are legally recognised platforms to conserve different subsets of those species. At the same time, we have clarified that our model contains species-relevant traits and processes but doesn't attempt to make exact species-specific predictions.	83–93, 211-214	I.8, M.10

It is unclear why the authors chose their habitat dataset, nor why they split it up into so many habitat types. If the intent is a broad-scale, multi-species perspective on PAs, I don't see how this approach really does this. I'd expect to see an analysis that runs on a map that integrates all land-cover types. If highlighting specific land-cover types is a key objective, that should be made explicit and explained.	We thank you for this pivotal comment, as it brought our attention to a strength of our study that was insufficiently explained. Our habitat dataset was chosen on the basis of policy relevance – specifically, priority habitats are listed (and thus mapped) based on (1) independent legal recognition and (2) importance for a further list of ~1,000 priority species. We now make this explicit in the text. We hope the referee can appreciate that we have taken care to integrate connectivity analysis into established legal structures for conservation. Still, we believe our general approach is readily applicable to other policy contexts or habitat definitions. We also recognise that while the habitat definitions we used are both legally and ecologically justified, a more representative set of nested habitat definitions would be possible in principle. As we stated in our discussion: “It is likely that species will make use of more than one habitat on a northward journey [...] networks of associated habitats would provide a more holistic assessment of connectivity.” Clearly we agree that an approach which integrates all land cover types, for each individual species, would be ground-breaking. However, unfortunately (1) we do not yet have enough data to take such a comprehensive approach and (2) the legal basis for conservation in England, and in the European Union, pertains to priority habitats as defined in the paper.	83–93	1.8
---	--	--------------	------------

Introduction			
38 – synergistic seems an overstatement in a general statement. Additive or coinciding may be true, impacts are not always interacting, multiplicative. Adding qualifying language like “Can” is likely more general and appropriate.	We agree. We now use “potentially synergistic”.	38	I.1
44-48 – repetitive with respect to the multi-generational statements. Condense.	We have made this version much more concise, while providing a clear explanation (requested by referee 2) that multi-generational connectivity is crucial in light of large spatial and temporal scales of climate change impacts.	44-52	I.3
50. Whilst – replace with synonym as meaning is unclear.	Done.	53	I.4
72- “as within the Maya.. to Berryessa project in CA”. Suggest you omit or elaborate to describe this project and how it demonstrates your point. There is too little information in the excerpt to be meaningful.	We decided to omit this in favour of explaining the policy context for the study, and our choice of habitat definition.	80-82	I.7

76-77 – For what kind of species have S-N shifts have been documented? This is a good place to indicate what kind of species your analysis will represent (or be relevant for) – plants, terrestrial animal, mammals, birds?	We now add broad details about the taxa for which species have, on average, exhibited S-N shifts. This falls within a new paragraph on the policy and ecological context of the study. This paragraph also gives details and references about the species the analysis is relevant for. These are a broad suite of specialised, legally protected species of plants, fungi, birds, beetles, butterflies, moths and more. Different subsets of these species are associated with the different legally protected priority habitat networks analysed. We have also added a note to the methods, linking our choice of a broad range of modelled dispersal abilities to the broad range of relevant taxa.	83–93, 211-214	I.8, M.10
78 – How is habitat defined, if a species-specific view is not used? Suggest omitting 'habitat' and replacing it with a more general term.	We hope that the new paragraph will give important context for the habitat definitions used. However, we also tried to clarify our wording to specifically highlight that we define 16 habitat networks based a national inventory of legally recognised, and ecologically-distinct, priority habitats.	83–93	I.8
82 – How is a patch defined?	We have added the detail that we consider the patch to be a contiguous clump of habitat. We do not go into more detail in the introduction, as we cover the specific handling of patches in the methods. Specifically, Condatis itself is run at a coarse scale (usually a 1km raster) but for all other analysis we define patches as contiguous clumps of 50m grid cells of a given habitat type based on a Moore neighbourhood (i.e. inclusive of diagonal adjacency between cells).	96-99, 101-102	I.9, I.11
Methods			

93-116. Move after methods/Condatis overview (or integrate within).	We now nest these sections beneath the Condatis overview.	120-217	M.1
153 – unclear.	We have now clarified that each patch was classified as ‘protected’ if more than 50% of its area was covered by PAs.	229-233	M.14
Results			
170. It is still unclear to me why you chose this dataset as representative of habitat. What about species that use more than one type of landcover? Why break this apart into types? This doesn't seem needed based on your stated research questions.	We hope we have addressed this in the paragraphs mentioned as a response to your similar opening point. In the text we now explicitly cover our rationale for using priority habitats. We also note that our approach has the benefit of being a conservative estimate of the connectivity of these landscapes. We acknowledge that including groups of habitat would be more ecologically realistic for more generalist species, but we are limited by our data, and legislation in England still pertains to these priority habitat types.	83–93	I.8
196 – Replace ‘approaches’ with a synonym.	We agree this was not the right word. Now replaced with ‘results’.	280	R.3
Discussion			
227 – omit: ‘we offer a cost-effective remedy’. This is overstated.	Now omitted.	318	D.3
228-239: The reads more like introduction than discussion.	We had intended this paragraph to reflect on the precedents for our specific findings, as well as possible explanations. We have reworded some of the paragraph to clarify this.	219-320	D.4
254 – if Aichi target is now defunct, why mention it in the introduction? Perhaps substitute another example.	We have removed this and replaced it with the upcoming Post-2020 Global Biodiversity Framework. Target 2 of the current zero draft includes reference to increase of effective and	346-347	D.5

	connected protected areas to 30% of the planet.		
Referee: 2			
I think this is an interesting idea and approach but as it stands there are some missing pieces.	We thank you for your time and interest in our study. We are pleased you appreciate our approach. Thanks to your clear and concise comments, we believe our paper now presents a more complete picture.		
It's not clear how north-south contiguity of habitat types by themselves will provide species with the ability to adapt to climate change, particularly when species are likely to move through multiple habitat types during the process of range expansion. This seems to be an analysis of the contiguity of habitat types, and thus akin to structural connectivity, but the authors haven't convinced me that this approach will be the best way to prioritize future protected area designation, as it lacks that linkage to species of conservation concern and/or climate-related range expansions across the region.	We are grateful to you and reviewer one for requesting a better explanation of the link between the chosen habitat types and conservation of priority species under climate change. We believe this has allowed us to highlight further strengths of our paper. We now include a paragraph with ecological and legislative justification for our focus on priority habitat networks, which clarifies the link to conservation of priority species under climate change. Specifically, we describe how the priority habitat networks are legally recognised, ecologically distinct platforms for conservation of subsets of a large number of priority species. We agree that range-expansions are usefully understood at the level of the species. However, a focus on habitats is justified because habitats are the stage for conservation, both in England and elsewhere. Habitats per se are protected and managed in protected areas, and ecological restoration takes place at the scale of habitats. We also agree that it is feasible that some priority species might move through multiple habitat types. We believe that an extension of our approach, integrating all habitat types for each individual species, would be truly ground-breaking. However, unfortunately (1) we do not yet have enough data to take such a comprehensive approach and (2) it remains that the legal basis for conservation in England, and in the	83–93	1.8

	European Union, pertains to priority habitats as defined in the paper.		
While I'm intrigued by the Condatis method, it's not clear why a circuit theory approach is best rather than just identifying contiguous habitat.	We agree that the use of Condatis should have been better justified (see responses above and below) but are not precisely sure what 'identifying contiguous habitat' means. Such an approach appears to assume that species would need a contiguous and unbroken tract of habitat to shift from their current range to new climatically suitable areas. This is clearly not the case – many species that are already shifting their ranges are moving through somewhat fragmented and non-contiguous habitat. The Condatis method allows us to approximate how a range shifting population would utilise habitat, as it moves over distances that may take many generations, and where, each generation, there is a chance of dispersing across regions of non-habitat 'matrix'. By doing so we can identify specific areas of habitat that are used by the population as it moves from source to target, which would be great places to protect. In landscapes where habitat happens to be arranged as contiguous 'corridors', Condatis will identify the benefits of this, but it is much more flexible besides – for example sufficient chains or 'sprinklings' of small stepping stones could amount to the same effect as a contiguous corridor. To reflect this in the text, we have now changed the wording of sentences in the introduction, and reworded the Condatis section of the methods. We hope this makes our choice in using this approach clearer to the reader.	83–93, 121-148	I.8, M.2
Also, Condatis is an uncommon approach and as such needs more explanation than what is provided for the reader to understand the theory, the output and how these differ from more	We have added detail to the Condatis section of the methods, explicitly stating how Condatis differs to other circuit theoretic models. Briefly, it differs in removal of the need for a resistance surface, and consideration	121-148	M.2

common approaches like Circuitscape and least-cost path	of colonisation between cells that are not adjacent to one another.		
Introduction			
Line 40: I would just delete the sentence starting with “Local management...” because it feels a little out of place and then allows you to connect your sentence about connectivity in conservation planning to how people study/or don’t study connectivity.	Done.	41-43	1.2
Line 46: “Demographic momentum for multi-generational range shifts” is not referenced and it’s not clear how this would happen. Do you mean the population will grow to a point where the likelihood of emigration increases?	Reviewer 1 also commented on these lines of the introduction. We have condensed these sentences to provide a clear explanation that multi-generational connectivity is crucial in light of spatial and temporal scales of climate change impacts.	44-52	1.3
Line 46-49: So, considering multi-generational connectivity is important because the effects of climate change will be slow or because species responses will be slow? Just more clarity here to understand the issue would be great.	Mainly because species responses will take place (and are taking place) over multiple generations. Most other connectivity metrics, including Circuitscape, implicitly assume that we want to quantify the areas that are ‘reachable’ by extant populations within one generation. This comment has also been addressed by the above change.	44-52	1.3
Line 67: The previous sentence implies that existing PA networks are enough to cover projected range expansion. If that is the case, then why would additional patches be critical for range expansion? It’s also not clear what ‘projection of patches’ means in this sentence. Do you mean projected range for expansion?	The studies we referenced assert that PAs facilitate range expansion due to observed increased colonisation rates - compared to non-protected patches - by the subset of species that are already shifting. However, they also note that this enhanced colonisation is likely due to the occurrence of higher quality habitat in PAs. These studies do not address the issue that an important subset of species are failing to shift their range currently, and that if protection was lost in critical locations, even more species could be vulnerable. Thus, although our wording may have caused confusion it can be simultaneously true that PAs are helping some species to shift and that more protection is needed.	67-76	1.5

	We have adjusted these sentences to add clarification. We think your second question was in relation to a typo, we have addressed that as well.		
Line 70: Can this be changed to “However there has been a significant delay in putting connectivity science into practice...” or rewrite that first part of the sentence to be more specific about what evidence and how long the lag can be?	Thanks - we agree that your suggested wording is better and have implemented it.	78-79	I.6
Line 72-73: The example at the end of the sentence comes without context or explanation. Can you provide more detail here so the reader can see how this fits?	We have now decided to omit this in favour of explaining the policy context for the study, and our choice of habitat definition.	80-82	I.7
Line 78: Explain what “high-flow” means here.	We go on to define flow in the methods. However, we agree that the mentioning of this term prior to its definition is potentially confusing and have therefore removed it.	101	I.10
Line 82: The difference between circuit theory in Condatis and other uses of circuit theory isn't quite clear. This is something that a lot of people will be interested in so I recommend you take another sentence or two to clarify. Is it just a conceptual difference – dealing with colonists versus dispersers? How is circuit theory used differently (maybe put in methods but more clarity would great).	We understand that this is an area we need to provide more detail. We have re-written the section in our methods regarding the Condatis methodology, and hope that it is now clearer how condatis differs from traditional circuit theoretic methods.	121-148	M.2
Last paragraph in Introduction: It's not clear what species or taxonomic group you're interested in here, which seems like a vital consideration given you're interested in long-distance, multi-generational connectivity for colonists and because the ability to track a changing climate also depends on	This has also been noted by reviewer 1. In response we have added a new paragraph into the introduction which includes the policies that lead us to our habitat choices, and includes reference to species groups in which south -north range shifts have been documented. Additionally, in our updated methods we now also clarify that we aren't	83-93, 211-214	I.8, M.10

species habitat requirements. If you're going for a species-agnostic approach it's important you state that with a brief explanation why and how doing that can still answer your question/inform conservation.	attempting to model specific species here.		
Methods			
Lines 99-100: I would replace shapefile format with vector format instead.	Done. We have also altered the wording of a preceding sentence and now use "polygon" and "vector" instead.	167,171	M.4
Lines 102-103: Can you be more specific about what "works best" means here? Perhaps something about the trade-off between resolution and memory. This could go into the supplemental info, but either way I think the readers would be interested in this.	We agree that this would be a good addition to the supplemental information. We have added a new appendix outlining the reasoning behind the need to find this equilibrium and a general rule of thumb for a standard desktop computer.	182	M.7
Lines 105-106: By "accurate measure" do you mean continuous value rather than categorical? I think it would be better to use those terms rather than accurate measure, unless you've gone out and ground-truthed the amount of forest cover and confirmed some level of accuracy.	We agree that different phrasing is needed here. We have altered it to explicitly state that we assigned 1km grid cells a value equalling the proportion of habitat coverage within it.	185-188	M.8
Line 113: Delete: "especially in the context of climate change."	Done	176	M.5
Line 116: Can you present the number of patches lost due to the rasterization – maybe with the proportion of area that these patches represent compared to other patches. That would really drive home that removal of these patches will have little consequence.	We agree that this would add reassurance that this issue was unlikely to affect our results. We have added a note on the mean proportion of area lost in this way in text, and added a column to supplementary table 1 outlining how much of each habitat was lost in this way.	179	M.6

Line 121: Can you elaborate on "... as if the network were an electrical circuit." I think you need a bit more detail before we can see how it's like an electrical circuit. (e.g., grid cells connected by resistors etc)	We have added to the Condatis section of the methods, which hopefully adds clarity on this point. Please see the sentence "In the Condatis analogy, the network is composed only of cells containing breeding habitat, a population entering the network from a source is considered to be the voltage, the time taken to colonise one cell from another equates to the resistance between the two, and the flow of the population colonising the available breeding habitat across those links is the current." within the enlarged Condatis methods section which we hope addresses this point	121-148	M.2
Lines 122-123: Can you elaborate on how both dispersal rate and reproductive rate can be used to parameterize resistance? Does a cell depict the value of one of those or both in combination? How do you obtain these values? Are they empirically estimated? And did you conduct any sensitivity analysis to these values?	In our changes to the methods section on Condatis, we now mention that the dispersal distance and reproductive rate are used to parameterise the distance-dependent dispersal kernel that defines the outcomes of modelled emigration events each generation from an occupied cell of breeding habitat.	121-148, 204-210, 212-215	M.2; see also M.9, M.10
Line 128: Can you clarify what you mean by this sentence? So, high connectivity value in a cell means high flow and high change in flow if that cell was removed from the landscape? Maybe you just need to clarify what "focal cell" is referring to?	We have altered this sentence to add clarity that flow represents how important each cell is to the overall connectivity of the network (a measure of which is provided by the conductance metric). We also now note that flow indicates the reduction in conductance you would expect to see if it was removed. We no longer use the term 'focal cell'.	121-148	M.2
Lines 138-139: Can you provide some explanation about why you chose 100 as the reproductive rate? Is this high or low, is it representative of a particular expectation, are you testing multiple values and then validating with observational data? Something similar to how you described the dispersal rate choice would be a great start. How are you varying reproductive rate by habitat type? I think you're not, but it's not clear then how resistance varies across the	We agree we should have stated clearly what our choice was based on. But the first thing to note is that we used the same value throughout and that the effect of R is to multiply up (or down) all flow and conductance values in proportion, so a different choice would not have affected any of our comparisons of interest. Because we aimed to compare the relative performance of networks and patches, rather than the absolute rates at which species might shift, we had no reason to choose R based on a particular species or group. Nevertheless, we	204-210	M.9

landscape. I'm wondering if you need a diagram to explain this...	would argue that 100 is well within the biologically plausible range. R is the number of emigrants produced per sq-km of suitable habitat per time unit, so 100 means that only one emigrant is produced per hectare. As such, if the time unit is considered as years, such a value might represent a medium-bodied vertebrate, or an invertebrate with a rather low population density.		
Line 143: What is meant by the largest gap in the network? This makes me wonder if you are examining the connectivity only of particular habitats, separately – so this is a structural connectivity analysis where you code grid cells of a particular habitat as having conductance and all other grid cells have no conductance? If this is the case, I think it would be helpful to understand how this could inform conservation, particularly when species are likely to move across different habitat types. Essentially, animals won't necessarily see a 'gap' in the network when moving from one patch to another when expanding their range. I guess I'm just not convinced that range expansion has to occur along a contiguous habitat type.	You are right in saying we are considering connectivity only of particular habitat classes. We have added a paragraph in the introduction which discusses the rationale behind our decision to do so and why, in this instance, we feel it is appropriate. We agree that animals don't necessarily see a 'gap' as it were. The Condatis methodology assumes that the population can move through the matrix, and this movement is captured by the circuit links that exist between habitat cells, even when they are not adjacent. We now explicitly state this in our methods - "The matrix between habitat is assumed to be homogeneous, through which the population can move, but cannot breed." As a point of clarification, conductance is a property of the entire network – e.g. a network of deciduous woodland habitat in England that potentially allows the northern border to be reached from the southern border. Conductance is not a property of individual cells. 'Flow' is a property of a habitat cell. Habitat cells are nodes in the network, and staging posts in the range expansion. A successful colonisation event must always start and end at a habitat cell, because only there can reproduction occur leading to the next generation. But this doesn't mean no movement can occur through the matrix, as hopefully clarified above. Our original aim was to not bog the reader down in too much of the technicalities, and direct them to the	121-148	M.2

	originating papers, if they were interested. We thank you for highlighting that, in doing so, our methods have become a little obtuse. We hope our new methods section is clearer in all respects.		
Line 148: What is a Moore neighborhood?	A Moore neighbourhood (sometimes called “queen” neighbourhoods) comprises a central cell and the eight cells surrounding it, it is one of two most commonly used neighbourhood types (the other being the von Neumann or “rook” neighbourhood which does not consider diagonal connections) To avoid confusion, we have now removed mention of Moore neighbourhood, and instead add clarification that in identifying contiguous clumps of habitat we included cells that share an edge and/or a vertex.	224-225	M.11
Paragraph starting at line 145: So really you are getting the fraction of flow for habitat patches that are smaller than the 1km²? Another diagram would be great here perhaps if that’s easier than explaining in the text, but that diagram could go in the supplemental info.	This is correct – while Condat is run at 1km², we apportion the flow of each 1km cell to discrete habitat patches within it. In this way we maintain relevance to decision making at the scale of the habitat patch. As requested, we have added an explanatory diagram to the supplementary material. The flow score for the 1km² is a result of the habitat contained within that cell; the cell value of the 1km rasters are equal to proportion they are covered by the habitat type in question. When we split the 1km flow score between habitat patches within the cell, we assign the amount of flow those patches likely contributed, as a result of their area. We have reworded some of the data preparation section, and the section you reference, to make our methods clearer.	186-189, 225-226, 227	M.8, M.12, M.13
Line 159: Do you mean flow when you say conductance? Or are you looking at the inverse of the resistance surface here? If it’s a different metric maybe you could repeat what it is here, and how it	Flow and conductance are two separate metrics. As noted above, conductance is a trait of the entire network, while flow is a trait of individual areas of habitat. There is no traditional resistance surface in	121-148	M.2

may differ from the typical definition of conductance in circuit theoretic approaches to connectivity modeling.	Condatis analysis; resistance in this context is derived from the distance between two cells and the dispersal ability of the population in question. In our regression analysis we are looking at how the connectivity of the whole network (summed up in a single figure called conductance), relates to the overall area of habitat networks. In our new methods section we explicitly define conductance and flow: "...circuit theory calculations lead to an evaluation of the overall connectedness of the network (defined by the metric 'conductance' - a property of the entire network), and the relative importance of each cell to the overall landscape connectivity (defined by the metric 'flow' - a property of individual habitat cells)." We hope that these alterations and additions address the confusion caused by our previous wording.		
Line 160: Have you defined what a habitat network is yet? Does that mean all patches of a particular habitat? I think you first mentioned this in the intro, but you need to make it clear what you mean by that given you have been mentioning patches and habitats separately as well. Maybe in this instance you just mean habitat type. This also comes up again in line 164 and I'm wondering if you mean habitat type here as well.	We have added clarification in the penultimate paragraph of the introduction that habitat in this context is based on a national inventory of legally recognised priority habitats, and that our 'networks' are the assemblage of patches that belong to a particular habitat type.	83-93, 96-99	I.8, I.9
Line 165: So low values signify reduced interior habitat, but reduced compared to what? Is it reduced compared to a completely contiguous patch?	Your understanding is correct. We have altered the wording of the section on GISfrag to better convey its meaning - "The degree of fragmentation of each habitat network was assessed using the GISfrag metric (40), where more contiguous patches, with large amounts of interior habitat, would have high values, representing a low degree of fragmentation."	244-245	M.15

Lines 166-168: It is not clear what you're doing here. I think we need a lot more detail.	We have added detail which hopefully addresses this point. We now state that the three different protection investment levels are “: a 1%, 10% and 25% increase in the proportion of each habitat that is protected” and add that “unprotected habitat patches were ranked by flow before being added to the PAs in descending order (highest flow first), until each of the three imagined protection investment levels were met.”	248-251	M.16
Line 170: Again, I feel like habitat network is strange here. It's a habitat type and you're just describing the area of that habitat type, but maybe is a network of patches of a particular habitat type and you're describing the area of a contiguous network of patches? I'm sure its just semantics, but right now I'm a bit confused.	You are absolutely correct that each habitat network in our study is a network of patches of a particular habitat type. We acknowledge that in the original format our choice of habitats and use of 'habitat network' may have been confusing. We hope that the new paragraph added to the introduction explains our choice of habitats and defines what we mean by habitat network. We also changed the mentioned sentence to read “The networks of priority habitat in England range...”	83-93, 96-99, 253	I.8, I.9, R.1
Other comments on methods: I think you should pull in some of the detail from your supp info regarding Condatis and circuit theory and description of conductance in this method to help clarify what you're doing.	We agree that clarity is needed on that section of the methods. We have brought in a lot of the descriptive wording from the supplementary materials, but have chosen to keep the mathematical equations in the supplements in order to remain as concise in the main text as possible.	121-148	M.2
Results			
Lines 175-183: This doesn't seem to be very surprising that more contiguous habitat have higher conductance and that larger patches are protected and more fragmentation occurs outside formal protection. Is there something more to it? Please do explain if there is!	There is not anything more to this, but this result for Conductance has not been demonstrated before in real landscapes. Furthermore, it sets an important context for our other results. Larger patches generally being more protected comes into play later on in our discussion, so we feel it is important to note it here.		
Line 191: Is blanket bog interesting enough to pull out here in the text? Is this an important habitat for threatened	Noted and removed.	275	R.2

species or climate change? If not you don't need to call it out.			
Lines 195-202: I'm wondering if it's appropriate to include both flow and area as predictors if they are correlated themselves and you are trying to interpret their coefficient value.	We agree that we should be cautious when including even slightly collinear predictors. Nevertheless, when checking the Variance Inflation Factor, a measure of how much larger the variance of the regression coefficient is than if the variable had been uncorrelated, for area and flow in each of the habitat networks we find a maximum VIF of 2.5 with all others are below 1.7. For reference, Multicollinearity becomes an issue when VIF nears 10. Furthermore, we would argue that flow and area being correlated, makes it all the more interesting that they have opposite relationships with protection. It indicates that, in a connectivity context, while we are protecting patches that should normally have more flow, we are still under protecting connectivity. This identifies a potential major blind spot in our designation system.		
Lines 203-213: This is interesting in that you don't have to protect a ton of area to get huge gains in flow protection (which is really structural connectivity). Going from 1-10% area has a much greater gain than going from 1—25%... But then it looks like this is contradicted in the last couple of sentences where area and flow are again tightly correlated (which makes sense!). Can this be explained more?	We are pleased that you appreciate the potential of our flow-led patch selection approach. We have rearranged these sentences to better convey what we meant by disproportionate (a median increase 57% connectivity due to a 25% increase in area protected is disproportionate). We have also removed reference to lowland dry acid grassland results in text, which was the source of confusion. This habitat is represented in figure 4a because it was the highest respondent to a 25% increase in protection, and was included in text by association (this was clarified in the figure legend, but we understand this was not received for some reason).	290-294	R.4
Other comments on results: There were no figure legends unfortunately, so it was difficult to understand your findings. I think you should move the supp info fig 1 into the main figures with maybe some habitat information and a	Our figure legends were uploaded in our initial submission, and we are not sure why reviewers couldn't see them but we thank them for performing the review despite this incompleteness. In the revision we have additionally	202, 573-586	

protected area layer as this will help clarify the methods and goals.	included the figures and legends in our main manuscript file. We have also moved supplementary figure 1 into the main document; it now includes the habitat data we used, colour coded as protected or unprotected.		
Discussion			
Lines 216-218: Can you provide any explanation why? Are these areas also valuable for human development whereas protected areas tend to be areas of less value to humans? I see you get to this later on, but maybe you should elevate this discussion to this first paragraph.	As you noted we address this in our discussion. Our opening paragraph is a summary of the core points of our manuscript, which we expand upon within the main body of the discussion. Implementing your suggestion would involve a wholesale rewriting of the discussion in order to maintain cohesiveness, which we feel is not necessary. We hope you can understand why we did not make this change.		
Line 222-223: While I agree with you in part, it is also because these habitats are undisturbed, which is a result of protection.	We agree that protection can mitigate habitat degradation, and have added this to the paragraph.	311-312	D.1
Line 224-226: I just want to point out that focusing solely on structural connectivity as you do in this study, rather than functional connectivity, and ignoring the connectedness of different habitat types in the role of range expansion is not exactly ecological realism either.	This is a salient point. We hope that we have gone some way to addressing this by explaining the rationale for our priority habitat focus in the new paragraph in the introduction. We agree that a more general approach would also be desirable, and certainly more ecologically realistic when modelling generalist species. However, we feel we have justified our approach both ecologically and legislatively. Not only is conservation legislation geared towards these specific habitat types, but this approach is conservative, focusing on those species that would find it hardest to shift-ranges, due to their constrained habitat requirements. Furthermore, increasing the connectivity of individual priority habitats also increases the connectedness of the wider habitats networks they are constituents of. Clearly action to facilitate range expansion for specialist species goes a	83-93, 379-381	I.8, D.6

	long way toward doing the same for generalist species.		
Lines 241-243: You show that contiguity of habitat type is lacking in many cases in the north-south direction and that many of these areas are not protected, but we don't actually see that these patches are important for range expansion or adaptation to climate change. Could you try and validate these models in some way using movement data or species distributions over time?	Supplementary figure 4 shows which patches in each of the 16 habitats studied would be important for populations of species with a variety of dispersal abilities to move from the south to the north of England. We acknowledge that this was not explicitly stated in the previous version of the manuscript and now state "We identify and quantify the important routes a wide variety of species are likely to take as they shift ranges in reaction to climate change, regardless of protection status (Fig. S3)." We provide evidence for our choice of axis of movement, and circuit theory (on which Condatis is based) has strong ecological and theoretical support. We agree that confirming our predictions through the use of real-world data, would be a natural follow on from this study. However, that is far beyond the scope of this study. Finally, there is the point that, by evaluating the landscape through the lens of range shifts we are inherently studying events that have yet to happen. In waiting to validate the outputs of these theoretic models we may miss our chance to facilitate the shifts that species need to make to maintain their range sizes under climate change. We now also mention this in text.	316-318, 387-390	D.2, D.7

Appendix C

Comment	Response	Line(s) in tracked changes document	Edit Code in tracked changes document
Referee: 1			
This is an improved version of the manuscript, with more detail and explanation.	We are glad that you think the previous changes improved the manuscript, and thank you for taking the time to review it for a second time; once again providing useful and thorough comments. We hope that the changes described below address your comments satisfactorily.		
My main comment is that I still don't think the authors adequately explained why the analysis is conducted for different habitat types, and why it makes sense to interpret large-scale connectivity in this way. The reader needs to understand not only the practical reason why England designates land this way (which the authors now partially explain), but more importantly why the analysis makes sense to conduct and interpret in this way. I appreciate that this is noted in the caveats section but I still wonder how useful the results are if they are not combined in some way to express overall connectivity of all patches and PAs – from both biological and planning perspectives.	We have made several additions to the MS related to this point, hoping to add clarity. A key change we have made which should address this concern is in the penultimate paragraph of the introduction: “We define habitat networks as assemblages of patches of a given priority habitat type, because priority habitats (1) receive distinct legal recognition and underpin planning decisions in England and (2) are highly ecologically distinct, providing for unique subsets of priority species.” Priority habitats are the reality of how habitats are recognised and protected, by law, in England, which we also now state this in the main text: “Some species depend on multiple priority habitats, but protection, restoration and conservation adequacy decisions are likely to consider each habitat individually” Ours is an applied analysis of connectivity, designed to be understood and used by conservation practitioners and policymakers. Thus, we stick to established habitat delineations to ensure that our analysis is relevant and useful given the specific legal context. However, we continue to assert (and clarify) that priority habitats are ecologically distinct and thus fit to be treated as separate habitat networks. Where we did not believe this to be	85-89, 98-101	I.6, I.8

	the case, we aggregated priority habitats (e.g. Lowland and Upland Calcareous Grassland).		
Needs a strong proofread for grammar, punctuation, and missing words.	Over the course of our edits we have given the manuscript a thorough proofread for grammatical errors		
Abstract			
23-24 – Awkward phrasing. Suggest: in fragmented landscapes, some patches are more important than others in maintaining population or habitat or landscape connectivity... or similar. I would omit the assertion that methods for identifying key patches are not widely used as it is difficult to assess the accuracy of this vague statement.	We now use your suggested phrasing, and have omitted the statement regarding the use of connectivity assessment methods.	23-26	A.1
25 – Few know what the Condatis methodology is. I suggest adding a brief statement that alludes to the type of modeling approach.	We agree that including some small description of the Condatis methodology here would be beneficial to the reader. We now include that: “we model range expansion through an adaptation of circuit theory”	26-27	A.2
28 – Remove ‘concerning.’ This is a value judgement. Replace with a number or objective description.	We have removed ‘concerning’ and now refer to 12 of 16 habitat networks for which flow protection falls short of area protection by on average 13.6%.	30-31	A.3
30 – consider changing ‘likely’ to ‘may be’ as you did not assess any other countries or the probability of having a similar situation, or reference as in 257.	We did not reference because Proceedings B does not include references in the Abstract. We have changed the wording to “may be” as you suggest.	33	A.4
34 – Remove ‘shows promising results’ as you don’t really evaluate whether this approach does better than other connectivity-based approaches. Suggest the message be re-framed as consideration of connectivity (using this method) provides an efficient means of identifying additional protected areas that prioritizes connectivity.	We have altered the wording to your suggested alternative. We now say: “... efficiently identifies additional protected areas that prioritise connectivity, protecting a median of 40.9% more connectivity in these landscapes with just 10% increase in area.”	36-40	A.5
Introduction			
54 – I don’t see this as a recent debate, just the re-emergence of the debate. Suggest removing ‘recent’ and briefly explaining what the new	We had not intended this to be interpreted as saying this is a wholly new debate, but understand that our wording could be read as such. We have removed the word ‘recent’, and	58-60	I.2

argument is.	now mention that the debate is in terms of species richness.		
55 – Need more words, citations, and qualifiers. Aggregated patches are better for many but not all species. What kind of habitat creation models are you referring to?	We have expanded and qualified, added one citation and clarified that we refer to a body of literature exploring metapopulation persistence and expansion under different habitat creation strategies.	61-65	I.3
64-65: Please add citations or state more speculatively.	We now use more speculative phrasing.	72-74	I.4
67 - This citation for recently putting connectivity science into practice is now a decade old, and there are now several examples of this being done. Suggest updating the tone message of this passage to suggest that there is a current movement to model and consider climate connectivity, but there is need for additional tools and methods to [insert a statement that defines your unique contribution].	Thanks for this suggestion – we now incorporated these changes.	75-80	I.5
72 – It would be helpful to add a statement that explicitly states why your analysis is divided out into different habitat types (and hence why your analysis is structured as such). What does ecologically distinct really mean in the context of your analysis? Does each type describe a unique subset of species with unique habitat requirements, that do not use other ecotypes? You have provided new content and explanations here and elsewhere, but I still don't understand why the analysis is conducted separately for each type.	We have added further clarification to this section. Most importantly, though, we add a statement in the penultimate paragraph of the introduction to clarify that our primary reason for using these habitat types is because this is the reality of how habitats are recognised and protected, by law, in our study region. However, we also justify our approach on the basis that the habitat types are indeed ecologically distinct (and we note that where the ecological distinction is not as clear, we have aggregated priority habitats into one network – e.g. Upland and Lowland Calcareous Grassland; see Data Preparation).	85-89, 98-101	I.6, I.8
78 – N-S shifts: Are these documented for your study area? If so, state this or where this comment applies to.	We now explicitly state that range-shifts in that direction have been widely documented in England.	92	I.7
88 – Is 'speed' the right word to use here? Seem in contradiction to circuit-theory interpretation that is more adept at describing movement resistance and landscape patterns, not speed.	Hodgson et al., 2012, showed that conductance is strongly correlated with the speed with which simulated metapopulations reach the target from the source. The methods we use are based on the work within Hodgson et al., 2012 and we therefore feel that speed is an apt word to use in this instance.		

	We hope that the addition to our methods now better explains the separation between this implementation of circuit theory and others, and therefore why some terminology that might not be appropriate for one may be appropriate for the other.		
Methods			
109 – I don't think circuit-based interpretations are limited to single movements by individuals. It represents movement as current through a landscape, with little consideration of actual movement distances. It doesn't always model steppingstone movement i.e., one patch to another (unless the analysis is explicitly set up to do so) – which is maybe more of what you mean to communicate? Please clarify.	Our intention wasn't to imply that other circuit theoretic methods could only be interpreted in terms of individuals. Rather we had meant that Condatis specifically includes the number of emigrants leaving a cell into its connectivity measures. We have removed reference to individual movements throughout the methods and the sentence now reads: "Unlike other uses of circuit theory Condatis models multi-generational movements by accounting for the production of emigrants in its assessment of connectivity." which we hope better conveys our point.	176-225, 180-182	M.1, M.3
120 – Please highlight that this analysis doesn't consider matrix heterogeneity that may affect accessibility of patches and functional connectivity of the landscape. This then prompts the question of how would your results changed if you considered matrix heterogeneity including barriers to movement? Accessibility is a big part of connectivity. Without this, the analysis seems to rely on distance and required movement from S to N, regardless of topography and other movement modifiers.	We now highlight that our simplified dispersal process is "a considerable assumption" but go on to give rationale for why we do so. Additionally, we now go further in explaining the differences between our application of circuit theory and traditional McRae based methods in the supplementary materials. We also mention in our limitations that by not considering the matrix, Condatis may "...overestimate the importance of some regions for those species that are hindered by landscape barriers."	195-197, 423-427	M.4, D.7
166 – Did you conduct a sensitivity analysis on your choice of 100 as R? How did you decide this was an appropriate value?	Reproductive rate (R), along with the proportion of habitat within a cell controls the number of dispersers leaving and arriving in a cell. Fecundity does not affect the distribution of flow, but instead affects the amount of flow and conductance proportionally across the whole network. As mentioned in text, our interest in this study was simply the performance of the networks and constituent patches, not the absolute rates of range-shifts. We did not conduct a sensitivity analysis for our	222-225	M.10

	choice of R, because the flow distribution (and therefore which patches were highlighted as the most important) would not change if we had chosen a different value of R. We decided this was an appropriate value because it represents a plausible “middle of the road”, reflecting the production of dispersers in the numbers you would expect for medium bodied vertebrates. We have added to the method text to make this point more clearly in the main text.		
Discussion			
262 - 263. If you are forcing current S to N without regard for matrix heterogeneity, I don't think you are really accounting for 'the important routes ... species are likely to take'. Suggest making this a more accurate statement. They are potential regions in which many species may move if tracking S-N shifts in the absence of movement constraints (e.g., topography), barriers etc.	We acknowledge that, without considering hard barriers to movement, Condatis may overestimate the importance of some regions – we now state that explicitly in the discussion. Furthermore, we do already mention that the inclusion of “climate refugia” data would be a good addition to our analysis to evaluate connectivity of those species not following S-N shifts. Nevertheless, we have altered the wording of this sentence, which now reads: “We identify the important routes a wide variety of species may take, using simplified dispersal assumptions, as they shift ranges from South to North in reaction to climate change, regardless of protection status”	331-334	D.2
Referee: 2			
The authors have done a really great job of making the requested changes to the manuscript.	We are grateful that you feel our initial revisions addressed the issues you raised in your previous review. We would like to thank you for taking the time to review our paper for a second time, and providing further insightful and useful suggestions. We hope that the changes described below will be received similarly. We note that there appears to be some discrepancy in the line numbers you have listed, and the line numbers in our manuscript. For instance, your final comment refers to line 452, while the text of our manuscript finishes on line 346. Unfortunately, the line numbers do not appear to		

	differ consistently, which has meant we aren't absolutely certain which line you refer to for some comments. We address all of your comments below, but ask for your understanding in the case of misattribution.		
However, they need to go further in their explanation of the Condatis method. They have stated that they preferred to reference literature rather than bog readers down in the details, and I agree, but they need a better balance because these methods are so uncommon and do not match perfectly to typical uses of circuit theory. Many readers will be coming with that typical use background. I outlined the biggest of these question marks in my detailed comments below.	We appreciate that we had not struck the right balance between accessibility and clarity within our methods section. However, we are limited in how much we can add intext due to the page limits enforced by Proceedings B. When contacting the editors, they recommended we focus on results and discussion and move methods to the supplementary materials where necessary. Nevertheless, we have made substantial changes to the methods in the main text, and added further detail to the supplementary materials regarding the differences between this implementation of circuit theory and others. Further detail of the changes made are noted alongside the relevant comment.		
The other big hole is the assumption that the matrix doesn't affect movement and that there is no variability in matrix effects on movement or breeding. I think there are three options here: (1) incorporate some piece of the matrix (e.g., road), (2) frame the study around structural connectivity (I think stepping stones can fit into this framing) and/or make it more clear what the study can and cannot do, and (3) hone in on several case study areas where roads and urban areas could block movement, as a description of the studies limitations or, better yet, next steps. These are just some ideas, but I think #2 is the best given the stage of the study. Is it possible to include any information about the matrix with this methodology? If so, this could be a future step. If not, this could be mentioned as a limitation.	We opted for (2) make it more clear what the study can and cannot do . We disagree that because of the assumption we make about the matrix our paper is purely a study of structural connectivity. Most measures have elements of functional and structural connectivity. We agree that the ideal situation would be to have parameters that exactly reflect the real world, but in modelling there is always a degree of disparity. We agree that not including movement barriers is an important assumption, and we now state that explicitly in text, but argue that our method allows the analysis of much more extensive networks without hitting computation limits. We also agree that we should be clearer regarding what our study can and cannot do. We hope that the additional text to the methods, discussion, and supplementary materials better emphasise this point.	176-225, 422-426	M.1, D.7
Comment regarding contiguous habitat: Though not the point of the comment, I agree it is definitely not the case that	In our more detailed methods section we now mention that the assumption of a homogeneous matrix is "a	176-225, 423-427	M.1, D.7

species need contiguous habitat to move across landscapes. The authors state this, then say that many species are shifting their ranges through fragmented, non-contiguous habitat. However, the Condatis method they have described does not account for the fragmented landscape outside of habitat cells of interest. Because of that, using circuit theory, areas of contiguous habitat will jump out as high flow. If the matrix were included, it is possible that some of these areas, stepping stones in particular, are actually cut off by roads of urban areas.	considerable assumption”, and explain our rationale for it. Additionally, we now also include this assumption, and the over-estimation of the importance of some regions it may lead to, in our limitations section. The inclusion of a resistance surface would involve a series of further assumptions about dispersal behaviour. The link between priority species and priority habitats used in our study is well established; to posit negative impacts of specific landscape elements on range expansion would be comparatively speculative (given the lack of data available).		
Comment about Condatis needing more explanation: See detailed comments below about the need for more information to make this methodology clearer. Removal of a resistance surface in particular needs explanation, as a circuit is needed in circuit theory, and a circuit contains resistors.	We hope that the additional text within the ‘Condatis Analysis’ section, in conjunction with our more detailed supplementary materials, better explain our methodology. In particular we now more clearly define how resistances are calculated. In the text we now define the dispersal kernel and explain how it is used to produce resistances and colonisation rates. Our supplementary materials now include much more detail on the Condatis method, as well as a table that explains how Condatis differs from Circuitscape-type methods.	176-225	M.1
Introduction			
85(54): Strict protection isn’t the only conservation intervention available for promoting/safeguarding connectivity, and often it is the less feasible option for conserving smaller patches in human-dominated landscapes.	We have changed this sentence to be in line with the below comment and be more flexible. It now refers to conservation rather than specific acts of restoration and protection.	55-56	I.1
133(101): This could be “targeted conservation” rather than protection to be more flexible than relying strictly on the addition of protected area.	Agreed we have changed it to “targeted conservation” as suggested.	114	I.9
Methods			
Line 143(111): Replace like with likely. Is there a reference for this sentence?	Done	179	M.2
145-147(113-115): Remove this sentence or explain it further either here or below. In particular, “consideration of	We have removed this sentence as requested.		

colonization events between non-neighbouring cells”			
148(113): This needs more information: How is resistance coded? Even though you say the need for a traditional resistance surface is removed, there certainly still is a resistance surface for which current can flow? Why are you assuming the matrix is homogenous, when many areas are impermeable to movement? Source nodes are located in the south; are those habitat cells? An equation to define how the dispersal kernel is parameterized is also needed as well as an explanation of how this affects the analysis in circuit theory terms? (I see some of this in the Appendix, but that is hard to follow) If the source nodes are located in the south and there is a dispersal distance that limits the size of the circuit, then how can the ground nodes be located in the north beyond this distance? I’m assuming there’s some iteration (the time-steps mentioned below?) but it is hard to follow.	We understand that in our attempts to not overwhelm the reader with mathematics, we did not find the correct balance. However, we must also be mindful that the manuscript doesn’t become so lengthy that it prohibits publication. Proceedings B has an upper limit of 10 pages, and the previous iteration of our MS was estimated to have reached this limit. Therefore, we are limited in what we can add to the text. Furthermore, having consulted with the editors they advised moving methods to the supplementary material to save on word count. With that in mind, we have made substantial changes to the methods section to find the desired balance: Where possible we have added to the ‘Condatis analysis’ section to answer your questions. Specifically, we explain the structure of the network in the Condatis analogy, and now explain that resistance is calculated via the colonisation rate calculated from the dispersal kernel, with a resistance link placed between every habitat cell and every other cell- which is how we circumvent the need for a traditional resistance surface from McRae style circuit theory. We also now include a definition of the dispersal kernel in text. However, the bulk of the specifics will still be in the supplementary materials. This has been added to extensively to better explain the Condatis analogy and how it is similar and different to traditional McRae style circuit theory (circuitscape), and we explain how Ohm’s and Kirchhoff’s laws are used to produce the Conductance and flow metrics. We hope that these changes better explain our methods and are at the same time easier to follow than the previous iteration.	176-225	M.1
160 (124): Remove “Given this definition”.	Removed.	205-206	M.5
161(125): Network of habitat patches or protected areas? How is conductance	We hope that the overhaul of the methods section explains this better, we were not able to move full	205-208	M.6

calculated? Is this total conductance across the circuit?	explanations of how conductance and flow are calculated, but now direct the reader to the correct appendix should they wish to see the underlying mathematical definition of these metrics.		
162(126): Importance of each habitat cell? What kind of cell? What is flow – current density?	Yes, we are referring to breeding habitat here. We hope that our changed methods, which includes a description of the structure of the network in the Condatis analogy better explains this point. ‘Flow’ in the Condatis method is the current flowing through a node,– and highlights more important areas for the eventual successful colonisation of the target (see new table S1). We mention in the supplementary materials that the current into a cell containing breeding habitat is the same as the current leaving it. To get Flow, we simply sum either the current in, or the current out.	176-225	M.1
165(129): Replace “countless” with “all” possible travel routes.	Done.	210	M.7
166(130): Follow the sentence starting with “Flow” with an explanation of what that means in practical terms – e.g., the degree to which connectivity would decline if that cell of habitat was lost.	Our new methods section and extended supplementary materials should now provide a full explanation of what flow is and how flow is calculated. We have added extensively to Appendix B the supplementary materials to better explain how Condatis calculations are made and what they mean, and our main text continues to state that “Flow is a good indicator of the reduction in connectivity that would occur if the habitat cell was deleted from the landscape”.	176-225	M.1
218-220(155-157): Are these values used as conductance (or the inverse as resistance)?	We think this comment refers to the sentence: “For each priority habitat network, each habitat cell was assigned a value equalling the proportional cover of the relevant habitat type within it (more habitat leading to both increased emigration and increased immigration).” We have removed this problematic sentence and have extensively rewritten and reordered the methods to add clarity on how the raster cells are used in Condatis. Specifically, we now explain the data preparation process first, allowing the reader to be introduced to the structure of our spatial data before explaining how	176-225	M.1

	they are used. We explain the Condatis analogy better and state: “In the Condatis analogy, each landscape cell containing breeding habitat becomes a node in the circuit network (cells from the rasters described in previous section).”		
221(159): There are some explanations of things in here that should follow directly below/within the section Condatis analysis. Can you combine? It is not clear how reproductive rate comes into the analysis. I see it in your dispersal kernel in the Appendix, but it needs some explanation in the main text, and this may be better suited in the second paragraph of Condatis Analysis.	Following your suggestion, we have restructured our methods section to flow better and hopefully be easier to follow. We have moved the definition of the dispersal kernel into the main text, which allows us to refer to specific parameters when describing reproductive rate, dispersal etc. helping the reader to understand how they are used in Condatis calculations.	176-225, 221-222	M.1, M.9
225(162): There’s some wording errors here.	Due to the issues with the line numbers we mentioned previously we were not able to identify the specific line you were refereeing to, so please forgive us if we have missed this point. However, given the extensive edits to the methods section (within which we think this refers) and further rigorous proof reading we hope that wording and grammatical errors have been caught and corrected		
252-253(167-168): Move what you have in parentheses to after the previous sentence mentioning R fixed at 100, so we know what that is supposed to mean.	Our changes to the ‘Condatis settings’ section should now better explain why we did not vary R in the way we did with dispersal distance.	221-222, 222-225	M.9, M.10
263-267(177-181): This is hard to follow with the wording, suggest removing “so we assigned flow from cells to patches”. The supplemental figure is really key to understanding this- great figure!	We have removed the portion of the sentence you suggested, and we are happy that the new supplemental figure provides the necessary clarity.	246-247	M.11
266(181): change 1km/2km to 1km or 2km.	Done.	248	M.12
308(208?): What does that mean, they were too few, sparsely distributed and fine in scale?	We believe this comment refers to the six habitat networks that could not be analysed as electrical circuits. If that is the case you are correct, the habitat was so sparsely distributed across England that Condatis was not able to ‘complete’ the circuit. One could interpret this to mean that a	278	R.1

	population with the dispersal abilities we included in this study could not move from source to target through this habitat. However, we stop short of making that statement. This is because a result such as this could be due to limitations in how computers handle floating points, called floating point arithmetic. When numbers get sufficiently small, which is likely when quantifying the rates of colonisation in sparsely spatially distributed habitat networks, a computer can no longer accurately represent them in memory and its ability to carry out calculations with them is compromised. This problem is called arithmetic underflow. Therefore, in cases like these six habitats, we cannot be certain that the reason Condatis can't complete the circuit is because species actually can't cross them, or if it is due to this limitation of computers. As such, when these instances arise it is easier to simply state that we cannot analyse them. We have added an explanation of floating-point arithmetic to the supplementary materials, to aid readers.		
342 (236?): Did you try a model with just flow?	We now include information on a GLM that uses flow as the sole predictor of protection status.	302-304	R.2
376(262): remove "exist to"	Removed.	331	D.1
414(282): Conductance gains less than the increase in area. Do you mean proportional increases in conductance and area?	Yes, we do mean proportional. We have edited the sentence to make this clear.	352-354	D.3
419(289): This study does not examine landscape permeability because it ignores the matrix. So maybe it's better to say safeguarding connected habitats and patches that act as step stones. The end of this sentence should be changed to connectivity rather than conductance; conservation planners are not discussing conductance usually.	We agree that "connectivity" would have been the correct word to use here and have altered the sentence as you suggested to reference connected landscapes and stepping-stones rather than landscape permeability.	358-359	D.4
420-423(294): The beginning of this sentence needs some rewording.	This sentence has been changed to "Patches that happen to be strategically located to act as South-North stepping stones may be small, and may lack other attributes that	364-367	D.5

	were important for past PA designation”		
424(296): “biased towards low-flow patches despite also being biased towards large patches with high flow” is contradictory and needs rewording and/or explanation. The last sentence in this paragraph is also confusing.	We have split this sentence for clarity. Patch size and patch protection have a positive relationship, but flow and protection have a negative relationship. This is interesting because, as we have shown, flow is positively correlated with area. Therefore, if large patches tend to be protected one would expect more flow to be protected. Flow having a negative relationship implies that we have protected the “wrong” patches for connectivity. We have now added additional sentences to clearly state this suggests a disconnect between past protection decision and connectivity. And mention that the presence of a size bias, and the similarity of protection decisions in other countries suggest that such a disconnect could also be present elsewhere. We have removed the final two sentences of this paragraph.	367-377	D.6
at 452(318): I think it’s a good idea to also mention that you didn’t examine the matrix. Some high flow areas may not be high flow at all if separated by massive urban areas and roads.	We agree that this is a major assumption in our study and have added a statement to that effect into this paragraph stating: “Next, the assumption of a homogeneous matrix may lead Condatis to overestimate the importance of some regions, for some species that are affected by physical barriers. However, by doing so the process of evaluating connectivity is made less computationally burdensome, itself a major limitation, whilst still maintaining the principles of isolation by resistance”	422-426	D.7
Figures			
Fig 2. Do you mean “proportion of protected total flow” here in (b)?	Yes. The wording used in the caption has been altered to: “. . .the proportion of total flow that is protected”		
Fig. 5 This is a hard to understand title. In (a), you say current conductance, but the y axis says total flow?	Yes, the references to conductance are remnants of a previous iteration of this figure that were not changed. The figure now refers to ‘Flow’.		

Appendix D

Comment	Response	Line(s) in tracked changes document	Edit Code in tracked changes document
Referee:1			
Thank you for addressing my previous comments and suggestions. I find this draft to be much improved. I have a few minor suggestions for further improvement.	We are pleased that you find our revised manuscript to be an improvement, we agree that it is much improved from its original form. We would like to extend our gratitude to you for taking the time to provide multiple thorough reviews, without which the improvements would not have been possible.		
Please edit to improve sentence structure and phrasing (e.g., 68).	We have re-reviewed the entire manuscript and have made a number of changes to sentences and phrasing.	Throughout	
Please re-consider the order of your discussion paragraphs or work on the transitions between ideas. I found this section a bit disjointed.	Thank you for this comment, which prompted us to re-evaluate our discussion. We agree that as the paragraphs were ordered our arguments and ideas seemed disjointed. We now that the reworked discussion more easily moves from point to point.	343-358	D.2
Introduction			
38-40. Rephrase.	We agree that this sentence could have flowed slightly better. It now reads: "Species can be hampered in their ability to shift ranges as an adaptation to climate change (1) where there are synergistic negative impacts of anthropogenic land use (2,3)."	38-41	I.1
45. omit 'furthermore'	Done	47	I.2
66. replace 'probably' with 'generally'	Done	70	I.3
75. 'form platforms'?	Replaced with "are platforms". In the UK, with some exceptions, protection and conservation is done through habitats rather than specific species. By "platform" we mean that through habitats 1000s of species are also afforded protection, and benefit from conservation. This is why we take a habitat approach, rather than tying to model specific species.	79	I.4

81. omit 'meanwhile during recent climatic warming', replace with 'Contemporary climatic warming'...	Done	85-86	I.5
85. rephrase.	We acknowledge that the previous wording of this sentence need work. It now reads: "In recent decades those species that have undertaken range-shifts have disproportionately colonised PAs, highlighting PA's key role in protecting habitats – even in species' potential future ranges"	88-91	I.6
Methods			
Methods are written in a mix of tenses.	We use the present tense for statements about the Condatis method and the data used that are true in general and not specific to this study. The past tense is used for methodological steps specific to this study. We have identified a number of sentences where the past tense needed to be used and modified their wording.	Throughout methods	
153. Clearly re-state here that unlike circuit-based analysis, a resistance surface/calculation only represents dispersal distances, and not the expected difficulty of moving through the landscape (as in circuitscape or omniscap). Because this information is excluded, movement barriers are not included.	We have added additional wording to this sentence, to add the further emphasis you request. It now states: "This simplified dispersal process is a considerable assumption. The way in which we calculate resistances does not model the expected difficulty of moving through the matrix, and means we cannot represent physical barriers to dispersal in the matrix as models like Circuitscape can"	159-161	M.1
163-170. Emphasize how the interpretation is different than simple circuit or least cost analyses. I.e., Connectivity from a multi-generational dispersal perspective; not landscape permeability due to movement ease/difficulty (without movement constraints as in traditional circuit analyses), or shortest path between 2 patches etc..	We now include that connectivity evaluations provided by conductance and flow are from a multi-generational dispersal perspective.	173	M.2
174 – what do you mean by traits and processes?	Traits and processes in this instance refer to the dispersal distances and reproductive rates that we use in our analysis (and to the very general ecological processes of reproduction inside breeding habitat and dispersal of propagules/offspring). We are not trying to make species specific predictions, but rather used values for dispersal and reproductive rate that are		

	representative for a broad range of species.		
Discussion			
270 – I’m not sure if you are talking about patches within a PA network or smaller patches within a larger patch. Rephrase.	We were referring to the fact that PAs mainly afford protection to habitats, rather than affecting species directly – therefore the emphasis on patches was slightly misleading. We have now reworded this to: “However, species will not directly respond to PA connectivity per se; it is the connectivity of the entire habitat network, whether or not protected, which affects the reproduction ...”	277-280	D.1